# Platform for isolation and characterization of SARS-CoV-2 variants enables rapid characterization of Omicron in Australia

Anupriya Aggarwal[1,13], Alberto Ospina Stella[1,13], Gregory Walker[2,13], Anouschka Akerman[1], Camille Esneau[3], Vanessa Milogiannakis[1], Deborah L. Burnett[4], Samantha McAllery[1], Mariana Ruiz Silva [1], Yonghui Lu[2], Charles S. P. Foster[2], Fabienne Brilot[5], Aleha Pillay[5], Sabastiaan Van Hal [6], Vennila Mathivanan[1], Christina Fichter[1], Andrea Kindinger[1], Alexandra Carey Hoppe[1], Mee Ling Munier [1], Supavadee Amatayakul-Chantler [7], Nathan Roth[8], Germano Coppola[7], Geoff P. Symonds[9], Peter Schofield[4], Jennifer Jackson[4], Helen Lenthall[4], Jake Y. Henry[4], Ohan Mazigi[4], Hans-Martin Jäck[10], Miles P. Davenport[1], David R. Darley [11], Gail V. Matthews[1,11], David S. Khoury[1], Deborah Cromer[1], Christopher C. Goodnow[4], Daniel Christ[4], Roselle Robosa [12], Damien J. Starck[12], Nathan W. Bartlett[2], William D. Rawlinson[3], Anthony D. Kelleher[1,11] and Stuart G. Turville [1]✉

Genetically distinct variants of severe acute respiratory syndrome coronavirus 2 (SARS-CoV-2) have emerged since the start of the COVID-19 pandemic. Over this period, we developed a rapid platform (R-20) for viral isolation and characterization using primary remnant diagnostic swabs. This, combined with quarantine testing and genomics surveillance, enabled the rapid isolation and characterization of all major SARS-CoV-2 variants circulating in Australia in 2021. Our platform facilitated viral variant isolation, rapid resolution of variant fitness using nasopharyngeal swabs and ranking of evasion of neutralizing antibodies. In late 2021, variant of concern Omicron (B1.1.529) emerged. Using our platform, we detected and characterized SARS-CoV-2 VOC Omicron. We show that Omicron effectively evades neutralization antibodies and has a different entry route that is TMPRSS2-independent. Our low-cost platform is available to all and can detect all variants of SARS-CoV-2 studied so far, with the main limitation being that our platform still requires appropriate biocontainment.

As of the end of 2021, severe acute respiratory syndrome coronavirus 2 (SARS-CoV-2), the causative agent of Coronavirus Disease 2019 (COVID-19), has accounted for close to half a billion infections and millions of deaths worldwide (https://covid19.who.int/). Throughout the pandemic, continuous viral spread has led to the accumulation of many polymorphisms primarily within its Spike glycoprotein. Many changes drove higher levels of transmission[1–4] and successive waves of infection where several variants of concern (VOC) dominated. In 2021, Delta VOC globally dominated, supplanting all VOCs Alpha, Beta and Gamma (https://nextstrain.org/ncov/open/global).

In Australia, the combination of lockdowns and international border closures led to a very low level of community cases during 2020[5]. In many states, the only appearance of the virus was primarily through quarantine networks for returning overseas travellers[6–8]. New South Wales (NSW) over this time welcomed the majority of returning travellers entering Australia and during the period of December 2020 through to June 2021, all VOCs and the majority of variants under investigation (VUI) were detected through rapid whole-genome sequencing (WGS)[9–11]. Access to this network through research-diagnostics partnerships enabled the study of emerging variants from geographically diverse sites at a time where community spread in Australia was either absent or minimal. In this setting, this partnership provided a sentinel programme, where genomic detection, isolation and characterization enabled observations which then informed on the risk of key emerging variants.

Previous work generating various cell lines and the powerful use of reverse genetics has helped many aspects of the COVID-19

[1]The Kirby Institute, University of New South Wales, Sydney, New South Wales, Australia. [2]Serology and Virology Division (SAViD), NSW Health Pathology, Sydney, New South Wales, Australia. [3]Hunter Medical Research Institute, University of Newcastle, Callaghan, New South Wales, Australia. [4]Garvan Institute of Medical Research, Sydney, New South Wales, Australia. [5]Brain Autoimmunity Group, Kids Neuroscience Centre, The Children's Hospital at Westmead, Faculty of Medicine and Health, School of Medical Sciences, Sydney University of Sydney, Sydney Institute for Infectious Diseases, Sydney, New South Wales, Australia. [6]Royal Prince Alfred Hospital, Sydney, New South Wales, Australia. [7]Department of Bioanalytical Sciences, Plasma Product Development, Research and Development, CSL Behring, Broadmeadows, Melbourne, Victoria, Australia. [8]Department of Bioanalytical Sciences, Plasma Product Development, Research and Development, CSL Behring AG, Bern, Switzerland. [9]Gene Therapy Research, Sydney, New South Wales, Australia. [10]University of Erlangen-Nürnberg, Erlangen, Germany. [11]St Vincent's Hospital, Sydney, New South Wales, Australia. [12]Molecular Diagnostic Medicine Laboratory, Sydpath, St Vincent's Hospital, Sydney, New South Wales, Australia. [13]These authors contributed equally: Anupriya Aggarwal, Alberto Ospina Stella, Gregory Walker. ✉e-mail: sturville@kirby.unsw.edu.au

response[12–14]. Alongside these approaches, our aim was to enable our sentinel programme by developing a rapid, accessible and scalable set of platforms that led to sensitive viral isolation and subsequent characterization. The utility of our platforms was then tested stringently at several levels. First, we tested our platform in rapid isolation of many globally relevant variants isolated from swabs with low viral loads (Quantitative Polymerase Chain Reaction Cycle Threshold (qPCR Ct) >30). Second, we characterized variants against a range of humoral responses and therapeutics. Finally, we demonstrated the utility of our methods to sensitively determine in vivo viral titres directly from nasopharyngeal swabs overnight. When the latter values are combined with diagnostic PCR, this enabled real-time in vivo measures of infectivity to particle ratio calculations (viral fitness). Following characterization of our platform, Omicron arrived in Australia in late November of 2021. In this latter setting, we rapidly isolated and extensively characterized Omicron. This highlighted that the solutions provided herein were valuable for the continued COVID-19 response, as they enabled the development of a scale of relative variant threat that combined measures of viral fitness with measurements of humoral/therapeutic evasion to inform relevant national and international guidelines.

## Results

**Rapid isolation of all VOCs and major VUIs through Australian quarantine.** In late 2020 and the first half of 2021, NSW hotel quarantine accommodated approximately half of all people entering Australia, with less than 1% being positive for SARS-CoV-2 infection (https://www.health.gov.au/sites/default/files/documents/2020/10/national-review-of-hotel-quarantine.pdf). As part of the COVID-19 response, rapid molecular surveillance was established with 3rd generation single-molecule sequencing technologies[11]. In this setting, we introduced a hyper-permissive HEK239T-based cell line that co-expresses ACE2 and TMPRSS2 (HAT-24), and was hyper-susceptible to rapid cytopathic effects (CPE) (Supplementary Fig. 1 and Video 1) compared with VeroE6-based cell lines. Susceptibility was based not only on the expression of physiologically relevant receptors ACE2 and TMPRSS2[15], but also on the selection of a clone that has revealed – by substantial dose-dependent CPE accumulating after 8 h post infection (Supplementary Video 1) and after 3 d in culture – sensitivities in viral detection/isolation approaching that of diagnostic PCR (Supplementary Figs. 1 and 2). The HEK293T line has limited innate viral immunity[16–20] and whilst the latter lack of immunity may contribute to the phenotype, there was the only clone that was hyper-permissive to infection (see Supplementary Fig. 1a–f for a representative clone of HAT-24 (vs HAT-10) in 6 contemporary variants).

The introduction of the hyper-permissive line HAT-24 enabled virus isolation from greater than 80% of positive swabs, with all VOCs (Alpha, Beta, Gamma and Delta) and 6 VUIs (Kappa, Eta, Zeta, Epsilon, Iota and Lambda) isolated successfully over a four-month period in early 2021 (Fig. 1a). Visual scoring of cultures revealed that many were unequivocally positive for CPE after overnight culture and had extensive CPE after 48 h (Fig. 1b–d), with a significant correlation ($r = -0.88$; $P = 0.02$) between average diagnostic PCR Ct values and titres (TCID$_{50}$ ml$^{-1}$) (Fig. 1a).

**R-20 is a rapid, high-content assay for live SARS-CoV-2.** Our next aim was to develop a fast, but low-cost, screening platform that could be scaled to high content. Using conventional approaches, we visually scored CPE to determine viral titres over 3 d to establish end-point titres as 50% Tissue Culture Infectious Dose (TCID$_{50}$) using the Spearman-Karber method[21]. In parallel, we used a high-content machine scoring method that enumerated the dose-dependent loss of 50% of nuclei (50% Lethal Infectious Dose (LD$_{50}$)) through live cell staining with Hoechst-33342.

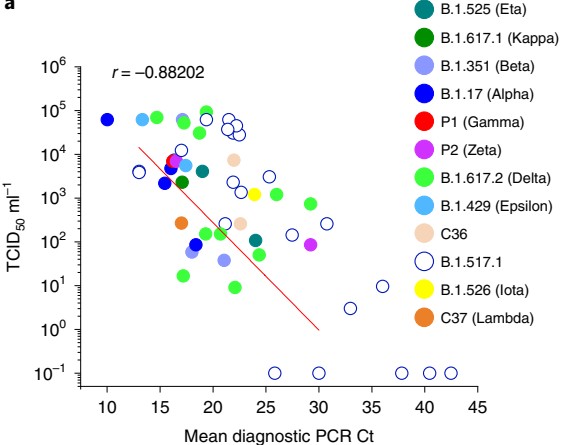

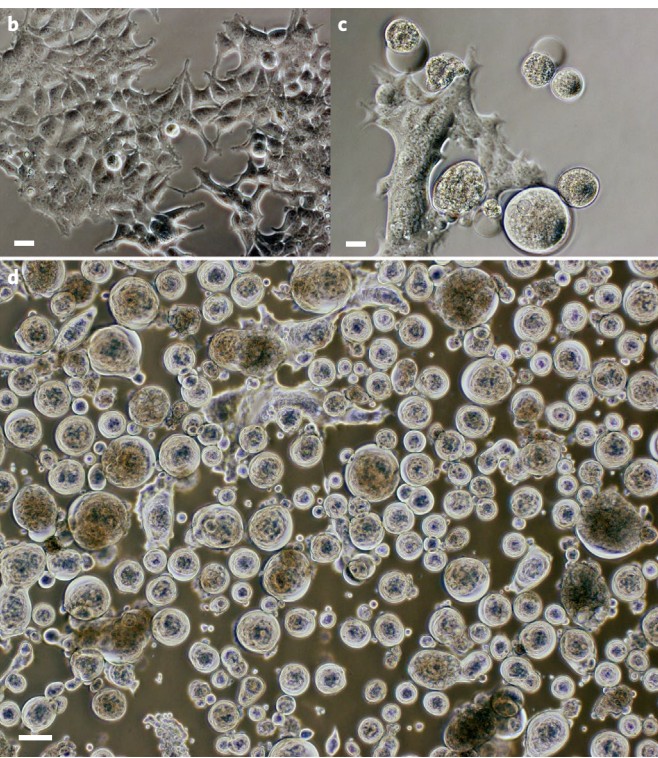

**Fig. 1 | Sensitive isolation of all SARS-CoV-2 VOC and key VUI over the period of February to May 2021. a**, Rapid end-point titres versus average Ct values (Ct values are averages of the Ct values across the viral genes detected (that is, primarily E, RdRP and N)) recorded using the HAT-24 cell line for rapid and sensitive isolation of SARS-CoV-2. Variant B.1.517.1 (open blue circles) is used herein for comparison, with samples derived from community spread in Australia from December 2020 to the end of January 2021 (n = 50). **b–d**, Rapid appearance of viral CPE (see large spherical viral syncytia) following overnight (**c**) versus 48 h (**d**) culture with a swab positive for the VOC Beta (**b** is mock uninfected control). CPE appearance is representative of all VOC. Scale bars in **b–d**, 10 μm.

Machine-driven scoring and counting compared the VeroE6 and the HAT-24 cell lines in two formats: after 20 h and after 3 d of culture. Analysis of the 20 h truncated format was only performed using the HAT-24 line, as the VeroE6 derived cell lines could not succumb to sufficient CPE after 20 h (Fig. 2a). Enumeration of nuclei in HAT-24 line 20 h post infection through high-content imaging/analysis revealed dose-dependent loss of cells, with only minimal variation between replicates (Fig. 2c–e). While VeroE6 cultures also revealed

similar dose-dependent CPE, this occurred after 3 d and was accompanied by greater variability (Fig. 2b).

End-point titres using the Spearman-Karber method were calculated at approximately $0.5 \times 10^6$ $TCID_{50}$ ml$^{-1}$ in VeroE6 and $0.5 \times 10^8$ $TCID_{50}$ ml$^{-1}$ in HAT-24 across all 12 variants tested (Fig. 2f,g). While we recognize the VeroE6 line to be relatively insensitive to infection, we further tested the more permissive VeroE6-TMPRSS2 cell line. The relative rank sensitivities of viral titres were HAT-24 > VeroE6-TMPRSS2 > VeroE6, with each increase in sensitivity being greater than an order of magnitude (Supplementary Fig. 1g).

Given the linearity of the dose-dependent curves, for stringency and elimination of operator bias we established titres on the basis of 50% loss of nuclei ($LD_{50}$)[22]. This was then correlated with traditional $TCID_{50}$ assays after 3 d of culture using the Spearman-Karber method (Spearman coefficient of correlation, $r = 0.62$, $P < 0.05$) across all variant titres tested. We subsequently refer to the overnight method in HAT-24 as R-20, a rapid high-content 20 h full virus screening platform.

We then cross-validated the R-20 platform using VeroE6s[23], given their availability and extensive use during the COVID-19 pandemic. For initial comparisons, we used the AB-3467 monoclonal antibody, as it is active against a broad range of SARS-CoV-2 variants[24]. We further cross-validated with the WHO international standard and reference panel for anti-SARS-CoV-2 antibody[25], in addition to the Plasma Alliance (https://www.cslbehring.com/newsroom/2020/covig19-plasma-alliance-expands-membership) internal serology standard. All cross-comparisons were carried out using 12 primary low-passage SARS-CoV-2 isolates. The R-20 platform observed all variants to be neutralized with similar $IC_{50}$ within the R-20 assay and in close correlation with that observed using VeroE6 (Fig. 2h–p). In addition, the R-20 platform revealed similar results to those laboratories that initially tested the WHO standard with live primary virus: 632 vs 686 for R-20 versus the mean contributing laboratories[25]. While R-20 is dependent on high-content microscopy and analysis, we further demonstrated that the R-20 assay is compatible with assays using standard optical multi-titre plate readers (Supplementary Fig. 3a,b).

**Immune evasion of SARS-CoV-2 at peak humoral responses.** For further stringent cross-validation, we turned to ranking of variants across a panel of sera from convalescent donors ($n = 24$) and mRNA vaccine (BNT162b2) recipients ($n = 24$). As observed with serological controls AB-3467 and WHO/Alliance standards, the R-20 platform produced similar results that were correlated with the longer yet more variable VeroE6 assays (Fig. 3a–d).

Using both assays, we also observed differences across vaccinated and convalescent cohorts. For instance, across all vaccine recipients with titres greater than 1/160, 19 had neutralization breadth that was similar across all variants with the exception of Beta (Supplementary Tables 1, 3 and 5). In contrast, the panel of convalescent donors derived from the ADAPT cohort highlighted a continuum of different phenotypes (Supplementary Tables 2, 4 and 6.). For instance, two donors (AD010 and AD016; Supplementary Fig. 4 and Table 2) maintained high titres and breadth across all variants, while one donor (AD062; Supplementary Fig. 4 and Table 2) had a potent response to early clade variants but could not reach end-point titres to the variants Epsilon, Delta, Kappa, Lambda, Gamma and Beta (Supplementary Table 2). With the exception of Gamma and Beta, all of these variants share the L452R or L452Q polymorphism in the Spike receptor-binding domain (RBD). In addition, while Zeta (D614G and E484K) was neutralized well across vaccine donors, it was significantly more evasive in convalescent responses (Fig. 3g–j). Overall, we observed Beta to be the most immune-evasive across both platforms, consistent with previous studies (Fig. 3g–j)[14,26–29].

**In vivo infectivity and fitness of community-circulating variants.** To establish the utility of the R-20 assay in determining viral infectivity from ex vivo samples, we curated two sets of specimens obtained during the acute phase of infection. In New South Wales, a quarantine breach at the end of 2020 led to singular seeding and community spread of the B.1.517.1 clade (Spike D614G and N501T). This clade provided primary samples with a genotype that was closely related to early circulating B clade D614G variant. The second cohort was derived during the month of July 2021 in the early stages of the Australian Delta outbreak. For each variant, samples were obtained at a time of no vaccination (B.1.517.1) or when 99% of reported cases (Delta) in NSW were transmissions in the unvaccinated (https://www.health.nsw.gov.au/Infectious/covid-19/Documents/covid-surveillance-report-20210813.pdf). In addition, given the low prevalence of COVID-19 in Australia, the majority also represent their first infection. While we could rapidly identify those individuals with high infectious viral loads using R-20, we also used the longer culture period for increased sensitivity (Fig. 4a–h). Using this strategy, we observed the sensitivity of 20 h vs 96 h cultures to be similar and their correlations with average Ct values to be similarly significant ($r = -0.8585$, $P < 0.0001$ and $r = -0.8606$, $P < 0.0001$, respectively). At both timepoints, virus could be detected at Ct values equal to and greater than 37 (Fig. 4h), thus approaching the sensitivity of diagnostic PCR assays.

**Fig. 2 | Titration and neutralization of SARS-CoV-2 viral stocks in VeroE6 and HAT-24 cell lines.** Virus stocks were serially diluted in 5-fold steps and added to cells in octuplicate. Cell nuclei were enumerated with high-content microscopy and cell numbers normalized to mock-infected controls where 100% represents cell numbers for mock-infected controls and 0% represents cell numbers for the highest viral concentration. **a**, Dose-dependent loss of nuclei in VeroE6-TMPRSS2, VeroE6 and HAT-24 cells at 20 h post infection (hpi). **b,c**, Titrations of a panel of SARS-CoV-2 isolates including the ancestral virus strain, VOC and VUI. Readout occurred at 72 hpi for VeroE6 (**b**) and 20 hpi for HAT-24 (**c**) cells. **d**, Representative fluorescence images of HAT-24 cells stained with Hoechst-33342, showing progressive loss of nuclei with increasing virus concentrations. **e**, Enumeration of stained nuclei with high-content imaging platform (IN Carta, Cytiva) using image thresholding and segmentation algorithms. Scale bars for **d** and **e**, 50 μm. **f,g**, Visual scoring of CPE at 72 hpi was done to calculate $TCID_{50}$ ml$^{-1}$ values in VeroE6 (**f**) and HAT-24 (**g**) cells (coloured bars). $LD_{50}$ values at 20 hpi are also displayed (small bars with pattern). Shown are the mean ± s.d. from at least $n = 3$ experiments. **h–p**, Neutralization assays were performed in high-throughput format with both VeroE6 and HAT-24 cells using live virus isolates from the VOC: Alpha (B.1.1.7), Beta (B.1.351), Gamma (P.1) and Delta (B.1.617.2), as well as the VUI: Epsilon (B.1.429), Zeta (P2), Eta (B.1.525), Kappa (B.1.617.1), Lambda (C.37) and C.36. 'Wildtype' virus from the same clade containing the dominant D614G mutation (Clade B - B.1.319) and ancestral Wuhan-like virus with the original D614 background (Clade A - A.2.2) were also included. **h,i**, Neutralization assay of SARS-CoV-2 isolates with monoclonal AB-3467 in HAT-24 (**h**) and VeroE6 cells (**i**). **j**, Spearman correlation of $IC_{50}$ values in **h** and **i** ($r = 0.9371$, $P < 0.0001$, two-tailed). **k,l**, Neutralization assay of SARS-CoV-2 isolates by WHO control sample G in HAT-24 (**k**) and VeroE6 cells (**l**). **m**, Spearman correlation of $IC_{50}$ values in **k** and **l** ($r = 0.8671$, $P = 0.0005$, two-tailed). **n,o**, Neutralization assay of SARS-CoV-2 isolates by the Plasma Alliance control sample in HAT-24 (**n**) and VeroE6 (**o**) cells. **p**, Spearman correlation of $IC_{50}$ values in **n** and **o** ($r = 0.7972$, $P = 0.0029$, two-tailed). Shown are the mean ± s.d. of technical replicates done in quadruplicate. Each panel is representative of a minimum of three independent experiments. In **h**, **i**, **k**, **l**, **n** and **o**, dose-response curves and interpolated $IC_{50}$ values were determined using the sigmoidal 4PL model of regression analysis in GraphPad Prism software version 9.1.2.

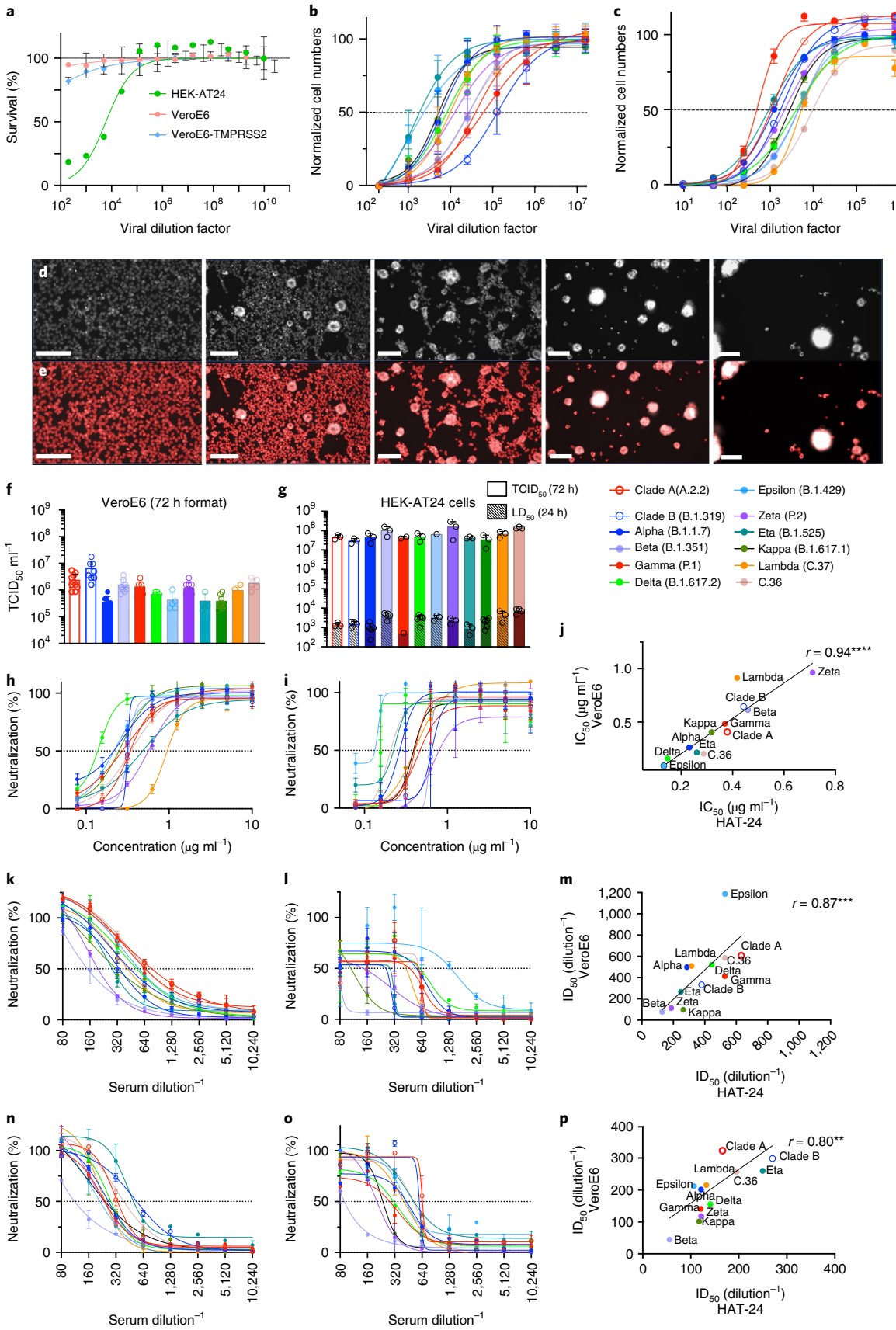

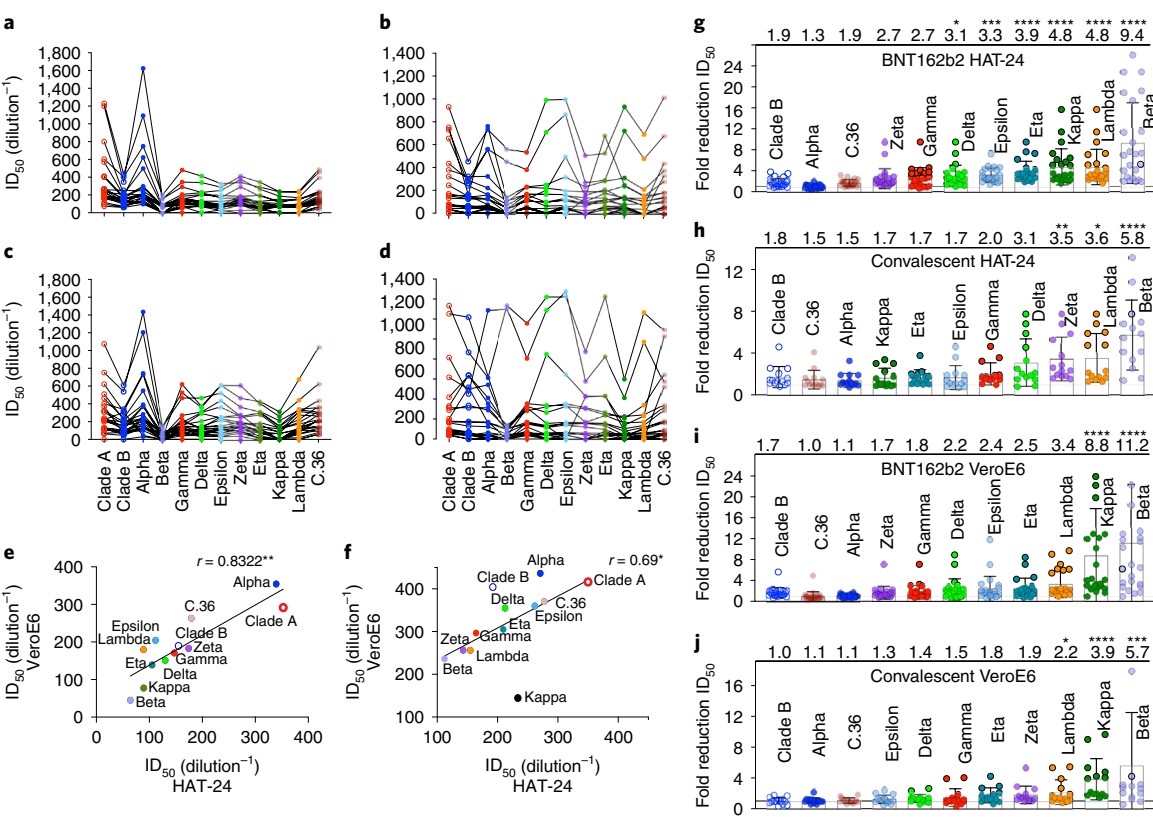

**Fig. 3 | Contemporary live SARS-CoV-2 variants and evasion of the vaccine and convalescent response. a–f,** Neutralization assays were performed in high-throughput format with both VeroE6 and HAT-24 cells using live virus isolates from the VOC: Alpha, Beta, Gamma and Delta, as well as the VUI: Epsilon, Zeta, Eta, Kappa, Lambda and C.36. 'Wildtype' virus from the same clade containing the dominant D614G mutation (Clade B - B.1.319) and ancestral Wuhan-like virus with the original D614 background (Clade A - A.2.2) were used as controls. **a–d,** ID$_{50}$ neutralization titres presented for 12 live variants for peak BNT162b2-vaccinated samples (**a,c**) (n = 24) and convalescent samples (**b,d**) for HAT-24 (**a,b**) and VeroE6 cells (**c,d**) (n = 23). **e,f,** Correlation of ID$_{50}$ for vaccine versus convalescent samples respectively for HAT-24 and VeroE6 cells. r = 0.8322 (P = 0.0013, two-tailed) for vaccine samples (**e**) and r = 0.6853 (P = 0.0170, two-tailed) for convalescent samples (**f**). **g–j,** Fold reduction in ID$_{50}$ values for data presented in **a–d** when compared to the earliest Clade A variant A.2.2. Shown are the mean ± s.d. for n = 24 donors in **g** and **i**, and n = 15 donors in **h** and **j**. Mean fold change for each variant is indicated at the top of the panel. Significance testing was done using Friedman's test with Dunn's multiple comparison. In **g**, P = 0.011 for Delta, P = 0.0001 for Epsilon and P < 0.0001 for Eta, Kappa and Lambda. In **h**, P < 0.0001 for Beta, P = 0.0015 for Zeta and P = 0.0105 for Lambda. In **i**, P = 0.0014 for Alpha, P = 0.0001 for Beta and P < 0.0001 for Kappa. In **j**, P = 0.0002 for Beta, P < 0.0001 for Kappa and P = 0.0230 for Lambda. Interpolated ID$_{50}$ values were determined using the sigmoidal 4PL model of regression analysis in GraphPad Prism software version 9.1.2.

Using linear regression (Fig. 4i), we calculated Delta to be 9.1-fold more infectious than B.1.517.1, for samples with equivalent levels of viral RNA (P = 0.0004). There was no correlation with high viral loads and infectivity across age groups (r = 0.10556, P = 0.3674) (Fig. 4j). To conclude, this analysis confirmed that primary swabs can be used to calculate end-point titres and combined with diagnostic PCR values to rapidly determine relative viral fitness. The primary limitation to this latter fitness analysis was accumulation of primary swabs from the community. To address a scenario when this is limiting, we investigated the relative fitness of in vitro isolated and expanded Alpha, Delta and Lambda variants (as Delta has circulated and supplanted both Alpha and Lambda globally). As the closest surrogate for in vivo expansion, we infected primary air–liquid interface (ALI) cultures[30] with the same multiplicity of infection (MOI) using culture-expanded viral preparations. Replication in primary cultures would then reveal greater fitness of each isolate upon testing the titres in an identical manner to that used in patient nasopharyngeal swabs. Titres from primary ALI cultures 3 d post infection were observed to resolve the fitness of each variant, with Delta>Lambda>Alpha (Fig. 5k), consistent with the known fitness within the community[31].

**Isolation and characterization of the first Australian Omicron case.**
The first case of Omicron in Australia was detected on 27 November 2021 and upon receipt of the primary nasopharyngeal swab, virus was expanded in the HAT-24 line within 48 h (Fig. 5a) and then further expanded for 2 d using the VeroE6 cell line. Initial attempts to use the VeroE6-TMPRSS2 line were not successful, as limited to undetectable titres were observed after 24 to 48 h of culture. Given the rapid spread of Omicron in the community, we were able to access many primary nasopharyngeal swabs to test for infectivity per diagnostic Ct value. In testing 34 Omicron primary samples, we observed a significant downward shift in the linear regression, a finding in direct contrast to its known transmission fitness in the community (Fig. 5b). An initial hypothesis for this result is a change in the mechanism of viral entry compared with all previous SARS-CoV-2 variants studied herein. The downward shift in linear regression in the HAT-24 line and limited replication in the VeroE6-TMPRSS2 line, are two observations that support Omicron being less dependent on TMPRSS2. To test this hypothesis, we examined the ability of Omicron versus Delta to be blocked by the TMPRSS2 inhibitor Nafamostat using the HAT-24 line. In this setting, we observed Omicron to be refractory to Nafamostat at the highest concentration tested. Therefore, we

conclude that the cellular entry pathway for Omicron has diverged from all variants before Delta and throughout the pandemic. As TMPRSS2 significantly increases SARS-CoV-2 entry for other variants (especially Delta), the downward shift in the linear regression would be expected with Omicron using the HAT-24 cell line. Given the increased transmissibility of Omicron within the community, the use of other proteases and/or cathepsins that are present in the upper respiratory tract[32] may have resulted in this change in cellular tropism (Fig. 5d). Importantly, the R-20 platform revealed Omicron's lack of TMPRSS2 use and can readily resolve whether future SARS-CoV-2 variants either continue along this path in tropism or switch back to TMPRSS2-mediated entry. Importantly, at this juncture it must be noted that any measure of viral fitness in any assay will need to consider all pre-Omicron variants as TMPRSS2-dependent, and hence rank their fitness accordingly. In Omicron and related lineages that no longer or poorly utilize TMPRSS2, a similar grouping based on tropism will need to be applied before any comparisons of relative fitness are made.

**Omicron resistance to humoral responses and immunotherapeutics.** Neutralization of Omicron using sera from donors with two doses of BNT162b2 or ChAdOx1 nCoV-19 vaccines only reached end-point titres at 1/20 in 4 out of 17 donors tested (Supplementary Table 7). To determine titre reductions to statistical significance, we selected high neutralization titre serum samples from the Australian ADAPT cohort[23] (source of serum summarized in schematic from Supplementary Fig. 5). Neutralizations performed with convalescent sera from early clade infections (Fig. 6a,b and Supplementary Fig. 5a,b) showed an average of 4.7-fold reduction for Beta, 1.5-fold for Delta and 2.2-fold for Gamma relative to the ancestral strain (A.2.2). All responses with Omicron were below the limit of detection (serum dilution 1/20). For convalescent sera obtained from Delta wave infections (Fig. 6c and Supplementary Fig. 5c), we observed a mean $ID_{50}$ of 52.6 for Omicron compared with mean $ID_{50}$ values of 770.5 for the ancestral strain, 211.1 for Beta, 317.5 for Gamma and 556.5 for Delta (Supplementary Table 8). Relative to the ancestral strain, this was a 20.7-fold reduction in neutralization with Omicron ($P < 0.0001$) compared with 4.0-fold for Beta ($P < 0.0024$), 1.6-fold for Delta and 2.9-fold for Gamma ($P < 0.0365$).

We next examined neutralization responses in sera from convalescent donors from early clade infections vaccinated with BNT162b2 or ChAdOx1 nCoV-19 vaccines (Fig. 6d,e and Supplementary Table 8). Here we again observe a 17.9 to 26.6-fold reduction in neutralization against Omicron ($P < 0.0001$) compared with 3.7 to 4.1-fold decrease observed for Beta ($P < 0.0028$). Similar results were observed with laboratory and healthcare worker volunteers at the peak of their third vaccine dose with BNT162b2, with a 16.9-fold reduction in neutralization against Omicron ($P < 0.0001$) compared with a 4.4-fold reduction for Beta ($P < 0.0001$) (Fig. 6f and Supplementary Table 9).

To observe neutralization responses at the population level, we then tested five polyclonal human IgG batches which comprise more than 10,000 pooled plasma donors collected during the peak of the US vaccine rollout (Fig. 6g and Supplementary Table 8). There was a 16.8-fold reduction in neutralization against Omicron ($P < 0.0001$) compared with a 3.3-fold decrease for Beta ($P = 0.0437$). Similar fold reductions were also observed from polyclonal IgG that was collected from convalescent donors between September and October 2020 (fold reduction of 14.1-fold for Omicron; $P < 0.0001$ versus 3.5-fold for Beta; $P < 0.05$) (Fig. 6g). Using the highest titre IgG batch (Poly_IgG – 1033) and one based solely on convalescent donors (Con-Poly_IgG – 869), we then tested all variants against Omicron to establish a ranking of humoral evasion to all dominant variants circulating in 2020 to 2021 (Fig. 6i,j).

We then determined activity across clinically available monoclonal antibodies (mAb) presently used in therapy. The class 3 antibody Sotrovimab, which targets a highly conserved region of the sarbecovirus RBD[33], retained neutralizing activity against Omicron. For other monoclonal antibodies, we observed either a lack of neutralization (Casirivimab, Imdevimab and Bamlaninvimab) or significant reduction thereof (Tixagevimab) (Table 1). These data were cited by a US Centers for Disease Control and Prevention (CDC) Health Advisory, forming updated recommendations for the use of monoclonal antibodies to treat COVID-19 when the Omicron BA1 lineage was accelerating in spread (https://emergency.cdc.gov/han/2021/han00461.asp).

**Omicron and implications for vaccine effectiveness.** We next estimated the fold reduction in neutralization for each variant within each cohort (with censoring). To do so, we grouped the data into three convalescent groups (Conv1-first wave, Conv2-second wave, Conv3-third wave (VOC Delta); see A, B and C in Supplementary Fig. 5), convalescent plus vaccinated, vaccine boosted (third dose), pooled polyclonal IgG and the WHO international reference standard as a control. The data are summarized in Fig. 6h. We have previously demonstrated that the neutralizing antibody titre is highly predictive of protection from symptomatic SARS-CoV-2 infection[34], and that the drop in titre to a given variant can be used to predict protection against the variant[35]. Using the protection curve in Khoury et al.[34], we estimated the level of protection for BNT162b2-vaccinated or boosted individuals in the first few months after vaccination (Table 2). This predicts a substantial drop in protection from symptomatic infection for BNT162b2-vaccinated and boosted individuals (compared with ancestral virus)[36], although with a relative preservation of protection from severe infection.

## Discussion

Our aim was to enable platforms to accelerate the COVID-19 response at two principal levels. First, the rapid and sensitive isolation of primary SARS-CoV-2 isolates down to low viral loads.

**Fig. 4 | Resolution of variant fitness in vivo by combining rapid and sensitive end-point viral titres with diagnostic PCR values. a**, Nasopharyngeal swabs were obtained from December 2020 to June 2021. In brief, samples were taken, placed into viral transport media and frozen at −80 °C in 100 μl aliquots within 24 h. Samples were filter-sterilized using 0.22 μm centrifugal filters and then co-cultured with the HAT-24 line. **b,c**, 20 h of culture: uninfected cells (**b**) versus infected cells (**c**) with a swab Ct <20. Note: CPE is scored through the rapid appearance of large spherical syncytia. **d,e**, 96 h of culture: uninfected confluent well (**d**) and low-level infection with CPE revealed as a combination of spherical syncytia and the formation of plaques (**e**). **f,g**, 96 h of culture: low-level infection with CPE revealed by extensive fusion across the viral cell sheet and the formation of plaques (**f**); extensive infection where CPE has resolved early as spherical syncytia and has become granular in appearance over time (**g**). **h**, Rapid end-point titres versus average Ct values as described in Fig. 1. Here the titres are scored within 20 h versus 96 h and correlated to the average Ct of all genes detected in diagnostic PCR values. All samples were collected as outlined in **a** and are from the Delta outbreak in Sydney in June 2021 ($n = 82$; $r = -0.8585$, $P < 0.0001$, two-tailed). **i**, Comparison of the early clade B.1.517.1 ($n = 15$) versus the VOC Delta ($n = 80$). Swabs for B.1.517.1 were collected as outlined in **a** but from December 2020 to January 2021. Spearman coefficient of correlation between end-point titres and Ct values, $r = -0.8582$ ($P < 0.0001$, two-tailed) and $r = -0.8543$ ($P < 0.0001$, two-tailed) for Delta and B.1.517.1, respectively. **j**, Correlation of Delta infectious titres versus age. $r = 0.1048$ ($P = 0.3709$, two-tailed). **k**, Titres derived from primary ALI cultures, 3 d and 7 d post infection using the R-20 platform. Shown are the mean ± s.d. from $n = 3$ experiments. Shaded areas in **h** and **i** represent 95% confidence intervals for each linear regression.

Second, enabling rapid characterization of their relative threat. The latter ranged from the ability to rapidly determine the evasion potential against a humoral response and/or therapeutics through to assessment of viral fitness in vivo and in vitro. For rapidly spreading pandemics such as COVID-19, the time taken for stringent resolution of a viral phenotype often lags. Time and precision are paramount in this context and the R-20 platform enables this at several levels, from time to virus isolation and expansion through to characterization.

Variants before Omicron all share similar changes within their Spike glycoprotein[1–4]. The changes are related to humoral evasion and viral fitness gains that have significantly increased transmissibility in variants such as Delta. While Beta was initially the most evasive variant in early 2021, Omicron emerged in late November as

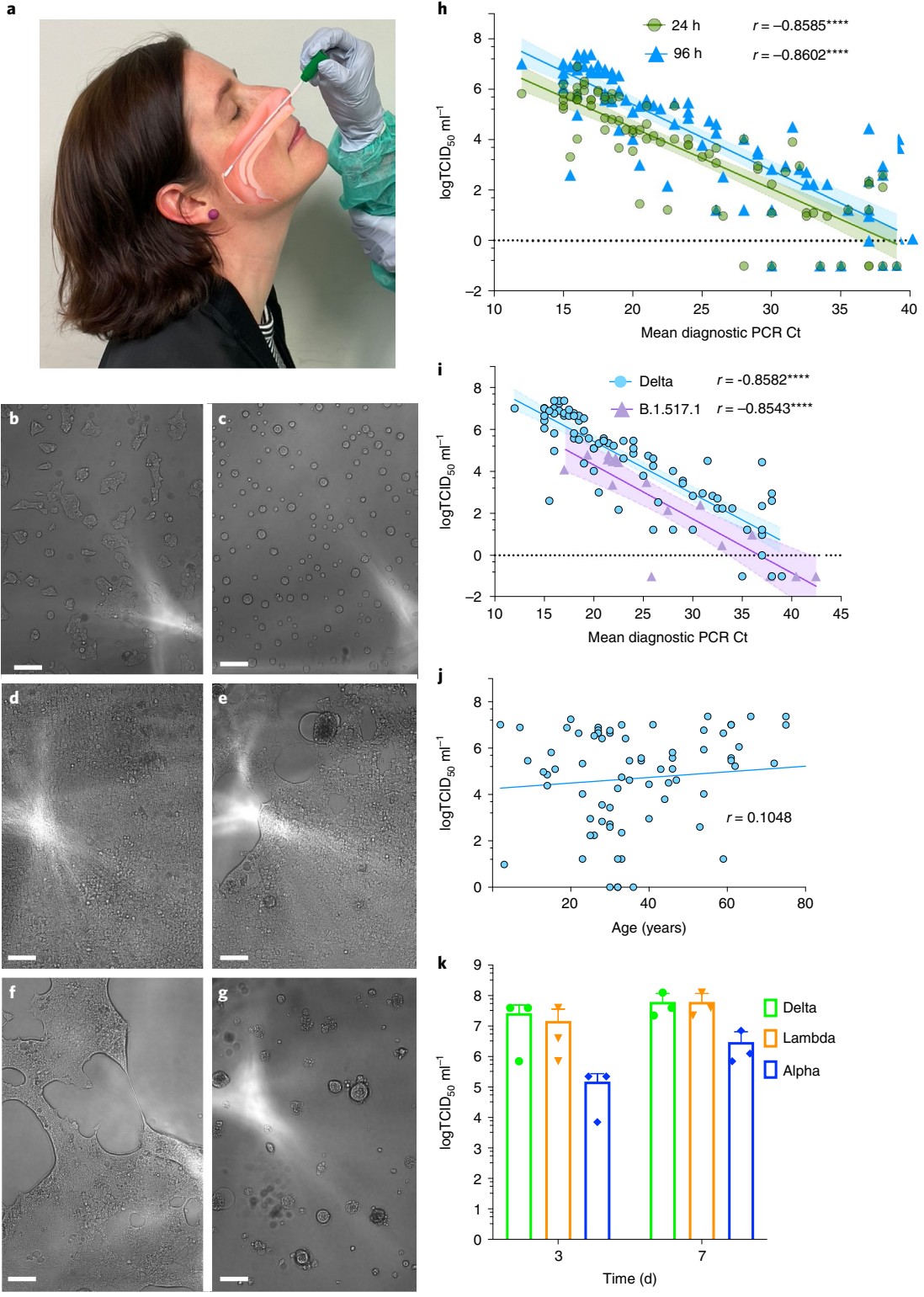

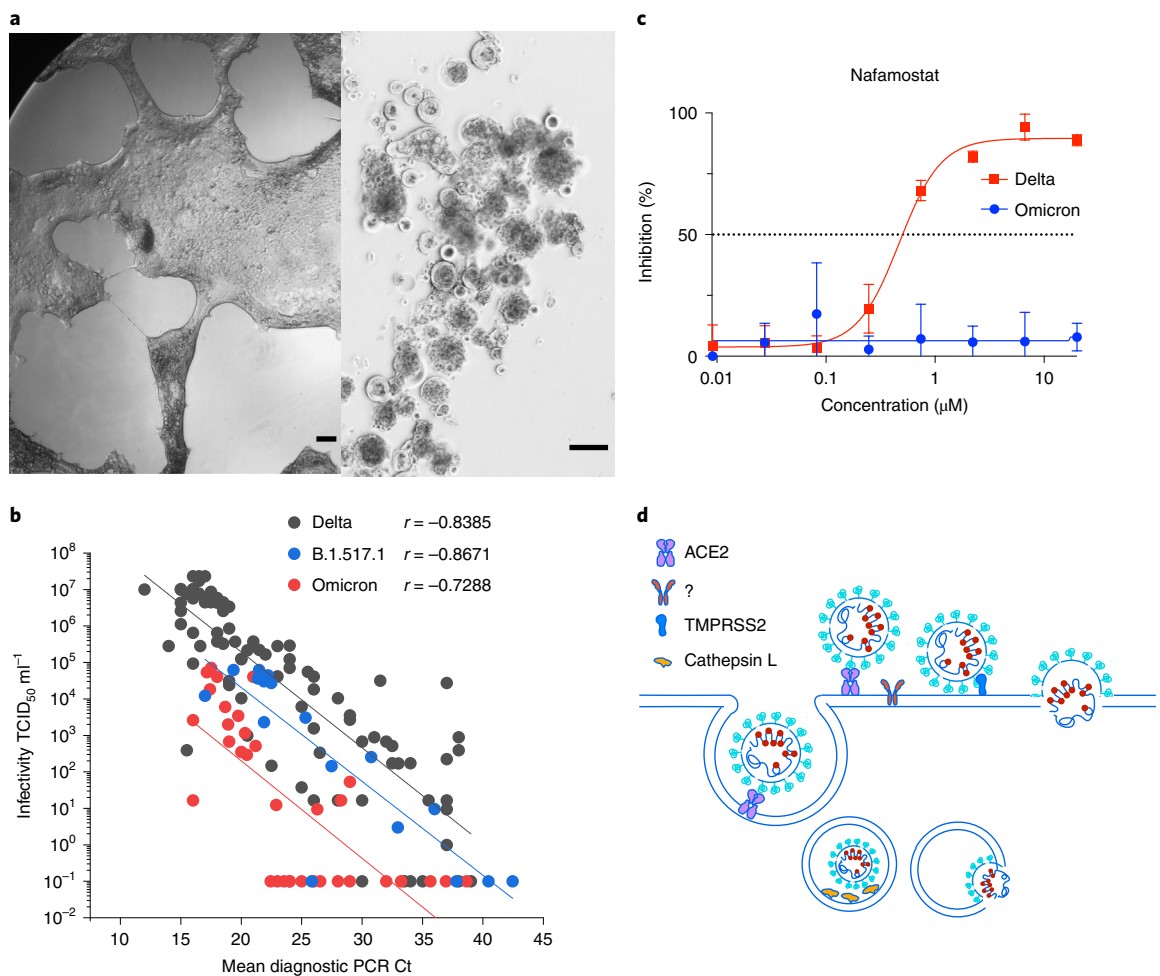

**Fig. 5 | Rapid isolation and characterization of the SARS-CoV-2 Omicron variant. a**, Primary Omicron sample (Ct 22) was filter-sterilized using 0.22 μm centrifugal filters and then co-cultured with the HAT-24 line. Left: 48 h of culture. Right: 72 h of culture. As with other SARS-CoV-2 variants, extensive syncytia accumulated within the HAT-24 line. Scale bars, 200 μm. **b**, End-point titres of Omicron plotted against the diagnostic PCR Ct value as described in Fig. 4. Spearman coefficient of correlation between end-point titres and Ct values for Omicron, $r = -0.7288$ ($P < 0.001$, two-tailed). Delta and the early variant B.1.517.1 linear regressions are overlaid for comparison. Data are from the 96 h timepoint. **c**, Titration of the TMPRSS2 inhibitor Nafamostat using the HAT-24 cell line. Note: the lack of inhibition in Omicron versus Delta at all concentrations tested reveals limited usage of TMPRSS2 by Omicron. Shown are the mean ± s.d. of technical replicates done in quadruplicate. Each panel is representative of a minimum of three independent experiments. **d**, Potential alternative pathways of viral entry in Omicron versus other SARS-CoV-2 variants. All SARS-CoV-2 variants can enter via endocytosis (left pathway) or at the plasma membrane by TMPRSS2 cleavage. The change in tropism of Omicron either reflects viral fusion primarily at the endosome using Cathepsin L or fusion at the membrane using an alternate protease to TMPRSS2 (designated herein as ?).

the most evasive and transmissible variant observed. Using the R-20 platform, we rapidly isolated then ranked Omicron across a continuum of humoral immunity in the community. Simultaneous to the submission of this work for review, many laboratories published pre-prints on the relative fold reduction of neutralization titres of Omicron[37–39]. As observed in the study of previous variants, the fold reductions covered a significant range. Importantly, it is imperative that we align actual vaccine efficacy with what we observe in vitro. Using data observed herein and an immunobridging modelling approach, we calculate a reduction in two-dose vaccine effectiveness to 37.2%, within the range of actual vaccine effectiveness observed in South Africa at 33% (https://www.discovery.co.za/corporate/health-insights-omicron-outbreak-analysis). While Omicron represented a significant challenge to the existing two-dose vaccination strategy, it fortunately has not translated to the mortality observed in unvaccinated communities in the early phases of the pandemic[40]. Infection in vaccinated populations is consistent with Omicron's ability to evade humoral immunity. Disease severity following

infection is multifactorial and would be tempered by both humoral and cellular immunity acquired through vaccination and/or previous infection. Yet not everyone in the community, such as the elderly or the immunocompromised, can establish a vaccine response that can lead to protection from severe disease. Fortunately for these at-risk groups, immunotherapeutic treatments such as Sotrovimab retain neutralizing activity for the Omicron lineage BA1. Unfortunately, preliminary data on the Omicron BA2 lineage have observed greater resistance to Sotrovimab (https://doi.org/10.1056/NEJMc2201933) and moving forward, therapeutics not targeting Spike, such as the protease inhibitor Paxlovid, may be more pragmatic for the treatment of at-risk groups against future circulating variants.

Viral loads based on diagnostic PCR with a variant in either unvaccinated or breakthrough infection[41–43] cannot often inform on the real risk of an individual's ability to spread a particular circulating variant or how previous vaccination may influence this. Furthermore, many people may be positive from diagnostic PCR but represent limited risk in spreading the virus in the community.

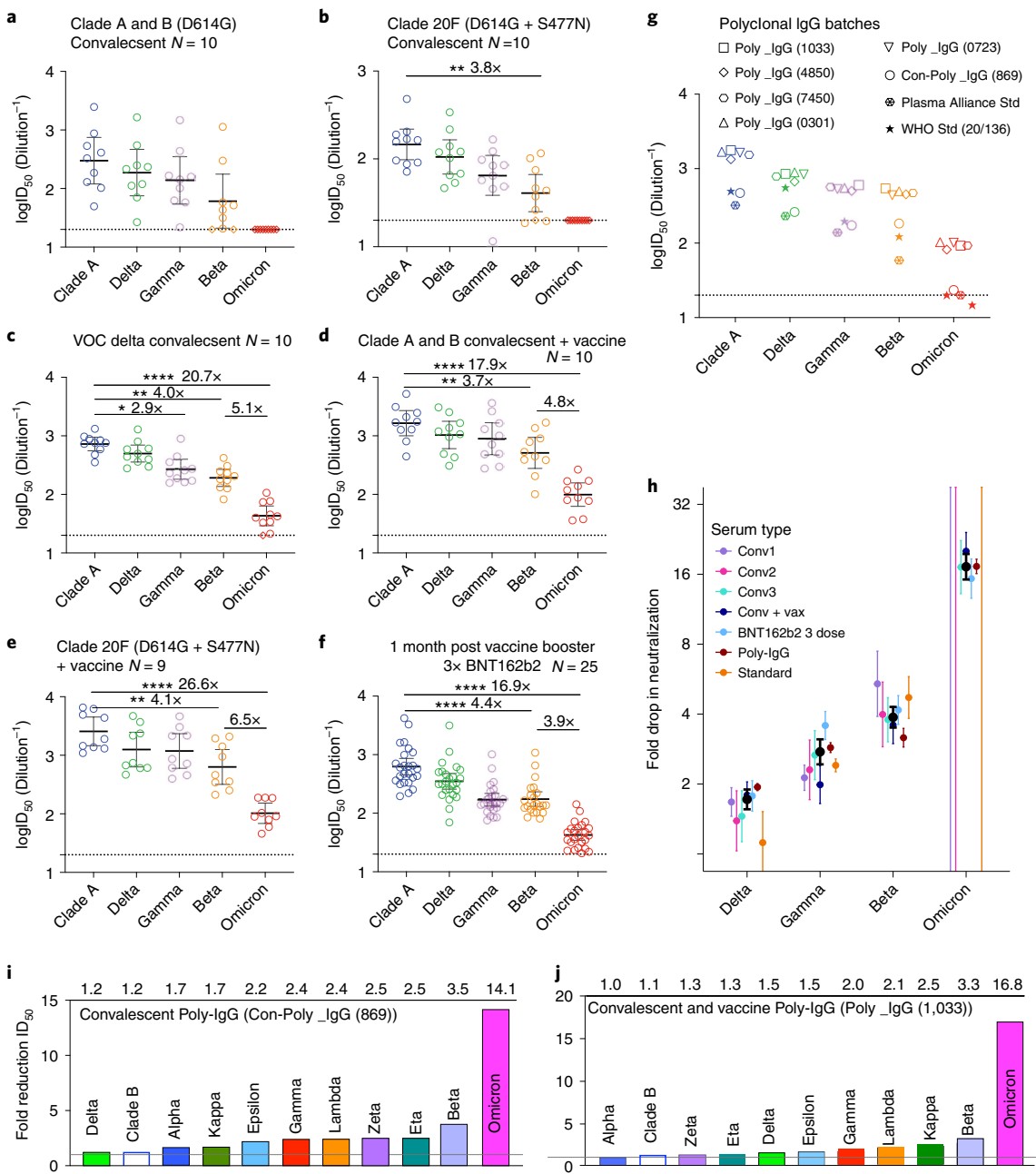

**Fig. 6 | Humoral neutralization of live SARS-CoV-2 variants in convalescent and vaccinated donors and concentrated human IgG plasma samples. a–c**, Neutralization assays were performed in a high-throughput format in HAT-24 cells using live virus isolates from the VOCs Delta (B.1.617.2), Gamma (P.1), Beta (B.1.351), Omicron (B.1.1.529) and the Clade A Wuhan-like virus, with the original D614 background (A.2.2) as a control. $ID_{50}$ neutralization titres are presented for five variants for convalescent donors from Clade A and B (B.1.319) (first wave) (**a**) ($n$=10), Clade 20F (D614G + S477N) (second wave) (**b**) ($n$=10) and third (VOC Delta) wave (**c**) ($n$=10). **d,e**, $ID_{50}$ neutralization titres presented for 5 live variants for vaccinated donors from Clade A and B (D614G) (first wave) (**d**) ($n$=10) and Clade 20F (D614G + S477N) (**e**) (second wave) ($n$=9). **F**, Sera from healthy donors one month post third dose of BNT162b2 vaccine ($n$=28). **G**, $ID_{50}$ neutralization titres across five live variants with concentrated polyclonal IgG from either convalescent and vaccinated donors or convalescent donors only. Data in **a–g** indicate the mean $ID_{50}$ of technical replicates for individual samples (circles) with the geometric mean and 95% confidence interval shown for each variant. Horizontal dotted lines in **a–g** represent the lowest serum dilution used and thus the limits of detection. Titres below the limit of detection are indicated with red diamonds. Fold-change reductions in $ID_{50}$ neutralization titres compare VOC to the ancestral variant, as well as Beta to Omicron. Significance testing was done using Kruskal Wallis test with Dunn's multiple comparison test. In **b**, $P$=0.0035 for ancestral versus Beta; in **c**, $P$=0.0024, 0.0365 and <0.0001 for ancestral versus Beta, Gamma and Omicron, respectively; in **d**, $P$=0.0028 and <0.0001 for ancestral versus Beta and Omicron, respectively; in **e**, $P$=0.0022 and <0.0001 for ancestral versus Beta and Omicron, respectively; in **f**, $P$<0.0001 for ancestral versus Beta and Omicron, respectively. **h**, Summary of neutralization reductions across Omicron and other VOC. Black dots represent the mean of all participants (except for Omicron where only groups of individuals with detectable neutralization against Omicron were included). Error bars are 95% CIs of the geometric mean. Unbounded error bars are present for groups where no detectable neutralization against Omicron was observed. **i**, Summary of fold reduction of neutralization potency across all variants tested using human polyclonal IgG derived from more than 20,000 US plasma donors (Poly_IgG-1033). This represents the collective activity of neutralizing antibodies derived from both convalescent and vaccinated plasma donors. **j**. Fold reduction of neutralization potency across all variants tested using human polyclonal IgG derived from COVID-19 convalescent donors' plasma[49]. Fold reduction values are representative of fold reduction from $n$=3 independent experiments.

**Table 1 | Neutralization of SARS-CoV-2 Omicron variant by commercially developed monoclonal antibodies and the class 4 Ab-3467**

| mAb | Developer | IC$_{50}$ (ng ml$^{-1}$) | | |
|---|---|---|---|---|
| | | Ancestral (A.2.2) | Omicron (B.1.1.529)[1] | Fold change |
| Sotrovimab | Vir Biotechnology / GSK | 372 | 1,059 | 2.8 |
| Casirivimab | Regeneron | 27 | nn (to 10 µg ml$^{-1}$) | N/A |
| Imdevimab | Regeneron | 25 | nn (to 10 µg ml$^{-1}$) | N/A |
| Bamlanivimab | AbCellera Biologics / Eli Lilly | 32 | nn (to 10 µg ml$^{-1}$) | N/A |
| Cilgavimab | Astra Zeneca | 18 | nn (to 10 µg ml$^{-1}$) | N/A |
| Tixagevimab | Astra Zeneca | 47 | 3,490 | 73.8 |
| AB-3467 | Burnett et al.[48] | 502 | nn (to 10 µg ml$^{-1}$) | N/A |

[1]nn, non-neutralizing at the highest concentration tested.

**Table 2 | Estimates of effectiveness (95% CIs)**

| | Omicron predicted effectiveness | |
|---|---|---|
| | BNT162b2 (2 dose) | mRNA boosted |
| Symptomatic | 40.8% (25.9%–55.0%) | 77.3% (65.1%–86.1%) |
| Severe* | 81.4% (50.9%–94.8%) | 96.4% (84.8%–99.3%) |

*Vaccine effectiveness against severe outcomes is much less well-validated as there is insufficient data available with which to parameterize the model at such low titres against severe outcomes, in addition to a lack of understanding of baseline severity with Omicron.

Through rapid end-point titres on the R-20 platform, we demonstrated infectious potential in an individual, and the relative infectivity per viral load between variants. This work also highlighted the inefficient use of TMPRSS2 and the divergence of the Omicron variant to another entry pathway. While preliminary studies support Omicron's cellular entry to be preferentially by endocytosis[32], an alternative entry pathway using ACE2 and a co-factor (for example protease) enriched in bronchial tissue would also be consistent with not only the observed shift in its tropism, but also the increase in its transmissibility. Importantly, if lower disease severity is linked to higher bronchial infection as observed in animal models[44,45], it is imperative that we continue to rapidly observe what entry pathways are used by emerging variants. If variants appear that again primarily use the ACE2-TMPRSS2 pathway, the platform herein can be readily used to resolve their appearance. This is similar to what was previously observed in other viral families, with the most documented being that for HIV-1 and its ability to chemokine receptor switch between CCR5 and CXCR4[46].

To conclude, the research herein provides an accessible, easy to use pipeline that enables resolution of SARS-CoV-2 variant threat at several important levels. First, in its ability to sensitively combine viral isolation, expansion and characterization of primary viral isolates from nasopharyngeal samples within one week. Second, in its ability to enable screening of thousands of compounds/samples reproducibly, efficiently, at low cost and at scale. While this was used in the resolution of the key phenotypes of many contemporary SARS-CoV-2 variants that emerged in 2021, its utility was tested with the emergence of Omicron at the conclusion of this study.

## Methods

**Ethics statement.** All human serum samples were obtained with written informed consent from the participants (2020/ETH00964; 2018/ETH00145; 2021/ETH00180). Primary bronchial epithelial cells (pBEC) were provided by P. A. B. Wark (University of Newcastle), and originally obtained from one healthy non-smoking donor (73-year-old female) during bronchoscopy, with written informed consent. Experiments were conducted with approval from the University of Newcastle Safety Committee (Safety REF# 25/2016 and R5/2017). All participants underwent fibre-optic bronchoscopy in accordance with standard guideline[47]. All primary isolates used in this study were obtained from de-identified remnant diagnostic swabs that had completed all diagnostic testing under approval by the New South Wales Chief Health Officer following independent scientific review (2021/NSWCHO H21/126831) and as outlined in the ADAPT ethics protocol (2020/ETH00964).

**Cell culture.** HEK293T cells (Thermo Fisher, R70007), HEK293T derivatives and VeroE6-TMPRSS2 (CellBank Australia, JCRB1819) were cultured in Dulbecco's Modified Eagle Medium (DMEM; Gibco, 11995073) with 10% fetal bovine serum (FBS) (Sigma, F423-500). VeroE6 cells (ATCC CRL-1586) were cultured in Minimum Essential Medium (MEM; Sigma, M4655) with 10% FBS and 1% penicillin-streptomycin (Gibco, 15140122). pBEC cultures were grown and differentiated until confluent in complete Bronchial Epithelial Cell Growth Basal Medium (Lonza, CC-3171) before use for air–liquid interface experiments. All cells were cultured and incubated at 37 °C, 5% CO$_2$ and >90% relative humidity, unless otherwise indicated.

**Participants and patient samples.** Three cohorts of human study participants were considered for studying neutralization of viral variants. The ADAPT cohort is composed of RT–qPCR-confirmed convalescent individuals recruited during 2020 in Australia[23], many of which have now been vaccinated. Of the >200 ADAPT participants, representative donors were chosen on the basis of serum neutralization titres against the Clade B virus in this study. A second cohort was composed of 24 healthy adult vaccine recipients who received the BNT162b2 vaccine in 2021. A third cohort was rapidly formed at the time of Omicron arrival in Australia and consisted of 25 healthcare workers who were 4 weeks post their third BNT162b2 vaccine dose. Median age, interquartile age range and female to male ratios are outlined for each group in Supplementary Tables 1–9.

**Monoclonal immunoglobulin products.** Monoclonal antibodies were provided by the Garvan Institute of Medical Research, Australia. Briefly, DNA sequences encoding the variable domain sequences of the therapeutic monoclonal antibodies Sotrovimab, Casirivimab, Imdevimab, Bamlanivimab, Cilgavimab and Tixagevimab were generated by gene synthesis, cloned into human IgG1 expression vectors and produced in ExpiCHO cells (Thermo Fisher, A29133)[48]. After production in ExpiCHO cells, monoclonal antibodies were characterized for binding to recombinant RBD by biolayer interferometry and for neutralization of live early clade (A2.2) SARS-CoV-2 virus[48].

**Polyclonal immunoglobulin preparations and anti-SARS-CoV-2 hyperimmune globulin.** A CoVIg-19 Plasma Alliance was formed in 2020 between major plasma pharmaceuticals including CSL, Takeda, Octapharma and Sanquin, with an aim to develop a COVID-19 immunoglobulin therapy. As part of that initiative, CSL Behring manufactured anti-SARS-CoV-2 hyperimmune globulin (CoVIg). Approximately 5,000 convalescent donor plasma units were collected between September and October 2020, exclusively from SARS-CoV-2 convalescent donors. After COVID-19 confirmation[49], the immunoglobulin was purified using the licensed and fully validated immunoglobulin manufacturing process used for Privigen[50], notionally similar to others[51]. Five intravenous immunoglobulin (IVIG) lots (Poly IgG 1033, 4850, 7450, 0301, 0723) manufactured using the Privigen process described by Stucki et al.[50] included US plasma collected by plasmapheresis from a mixture of donors vaccinated with SARS-CoV-2 mRNA vaccines, convalescent and non-convalescent donors (source plasma, n between 9,495–23,667 per batch). The majority of donations were collected between April and June 2021 (Supplementary Fig. 5). The Plasma Alliance control was obtained from CSL Behring and the WHO international reference standard for SARS-CoV-2 neutralization (NIBSC 20/136) was obtained from NIBSC[25].

**Generation of HEK293T-ACE2-TMPRSS2 cells (clone HAT-24).** HEK293T cells stably expressing human ACE2 and TMPRSS2 were generated by lentiviral transductions as previously described[23]. Briefly, the open reading frames for hACE2 (Addgene, 1786) and hTMPRSS2a (IDT, synthetic gene fragment) were cloned into lentiviral expression vectors pRRLsinPPT.CMV.GFP.WPRE[52] and pLVX-IRES-ZsGreen (Clontech, 632187), respectively. For ACE2 cloning, Age1/Bsrg1 cut sites were used to replace GFP with ACE2, while hTMPRSS2a was cloned into pLVX-IRES-ZsGreen using EcoR1/XhoI restriction sites. Lentiviral particles expressing the above genes were produced by co-transfecting expression plasmids

individually with a 2nd generation lentiviral packaging construct psPAX2 (courtesy of Dr Didier Trono through NIH AIDS repository) and VSVG plasmid pMD2.G (Addgene, 2259) in HEK293T producer cells using polyethyleneimine as previously described[53]. Virus supernatant was collected 72 h post-transfection, pre-cleared of cellular debris and centrifuged at $28,000 \times g$ for 90 min at 4 °C to generate concentrated virus stocks. Two successive rounds of lentiviral transductions were then performed on HEK293T cells to generate ACE2-TMPRSS2 HEK293T cells. Clonal selection led to the identification of a highly permissive clone (HAT-24), which was then used in subsequent experiments[23].

**Viral isolation from primary specimens.** Respiratory specimens were collected and stored at 4 °C for same-day diagnostic RT–qPCR (Allplex SARS-CoV-2 Assay, Seegene). Specimens positive for SARS-CoV-2 were then frozen at −80 °C within 24 h of collection, and later transported to a certified BSL-3 facility for primary isolate propagation. Thawed viral eluate was sterile-filtered through 0.22 µm column filters (Merck, UFC30GVOS) at $10,000 \times g$ for 5 min and then serially diluted (3-fold series) in quadruplicate. Viral dilutions were added to HAT-24 cells seeded in 96-well plates at $10^4$ cells per well (final volume, 100 µl). Plates were incubated at 37 °C and monitored by brightfield microscopy every 24 h using high-content microscopy. Once extensive CPE became evident in at least 2 dilutions (Fig. 1e); the cells and supernatant from these cultures were collected and cleared from debris by centrifugation at $2,000 \times g$ for 5 min, aliquoted and stored at −80 °C (passage 1). For further expansion, 300 µl of passage 1 virus including infected cells was used to resuspend a pellet of $0.5 \times 10^6$ VeroE6 cells in suspension. After incubation for 30 min at 37 °C, the co-culture was transferred to a 6-well plate with 2 ml of MEM-2% FBS medium per well. The resulting supernatant was collected after 48 to 72 h (when visible and extensive CPE was observed), cleared by centrifugation as above and stored at −80 °C (passage 2). A final larger expansion step (passage 3) was conducted by resuspending $2 \times 10^7$ VeroE6 cells in 500 µl of diluted passage 2 virus (MOI = 0.05), incubating at 37 °C for 48 h, and clearing and storing the supernatant at −80 °C as above. Shorter expansions with higher titres were also achieved in pre-Omicron variants by infecting the VeroE6-TMPRSS2 cell line with the same MOI but collecting cells at 24 h post infection. Sequence identity and integrity were confirmed for both passages 1 and 3 virus via whole-genome viral sequencing as described further below.

**Titration of primary nasopharyngeal swabs.** Viral eluates from primary specimens were thawed and sterile-filtered as indicated above, and coded before leaving the diagnostic laboratory. Samples were then diluted in 96-well plates (3-fold series, in quadruplicate) and 40 µl of each dilution were transferred to an equal volume of freshly plated HAT-24 cells seeded in 384-well plates (CLS3985, Corning) at $1.6 \times 10^4$ cells per well in DMEM-5% FBS medium. After incubation at 37 °C for 72–96 h, whole wells were imaged by high-content brightfield microscopy, and images were binarily scored by two independent experienced operators for CPE (+ or −) to determine the viral end-point titre (that is, last dilution containing at least one '+' well).

**Whole-genome viral sequencing.** Clinical respiratory specimens positive by diagnostic SARS-CoV-2 PCR were sequenced using a combination or Nanopore single-molecule sequencing and amplicon-based Illumina sequencing approach, as previously described[11]. Consensus SARS-CoV-2 genomes have been uploaded to GISAID (www.gisaid.org) and are publicly available as indicated in Supplementary Table 1.

**Titration of expanded viral stocks.** For overnight titrations (20 h format), HAT-24 cells were trypsinized, resuspended in DMEM-5% FBS medium with Hoechst-33342 live nuclear dye (Invitrogen, R37605) at 5% v/v and seeded in 384-well plates (Corning, CLS3985) at $1.6 \times 10^4$ cells per well. For traditional titrations (72 h format), HAT-24 or VeroE6 cells were seeded at $5 \times 10^3$ cells per well in DMEM-5% FBS or MEM-2% FBS, respectively, and stained with Hoechst-33342 only after the 72 h virus incubation. SARS-CoV-2 viral stocks were serially diluted (5-fold series) in cell culture medium in octuplicate and then 40 µl of viral dilution were added to an equal volume of the freshly plated cells. Plates were incubated for either 20 h (rapid overnight titration) or 72 h (traditional method) before the entire plate area was imaged on an InCell Analyzer HS2500 high-content microscope (Cytiva). Brightfield images were visually inspected by two independent experienced operators and compared against negative and positive infection controls to score wells binarily for CPE (+ or −) to calculate $TCID_{50}$ values according to the Spearman-Karber method[21]. Fluorescence images were processed with IN Carta analysis software (Cytiva) to obtain total nuclei counts per well. For calculation of $LD_{50}$ values, cell counts were normalized so that 100% represents the average cell number for mock-infected controls and 0% represents the average cell number for the highest viral concentration tested. $LD_{50}$ values were obtained with GraphPad Prism software using the nonlinear regression for dose-response with variable slope and four parameters. Viral titration on VeroE6 cells was performed exactly as described above, except that cells were seeded at $5 \times 10^3$ cells per well in MEM-2% FBS medium and stained with Hoechst-33342 after a 72 h incubation.

**Rapid high-content SARS-CoV-2 microneutralization assay with HAT-24 cells (overnight R-20 assay).** HAT-24 cells were trypsinized, resuspended in DMEM-5% FBS medium with Hoechst-33342 live nuclear dye (Invitrogen, R37605) at 5% v/v and seeded in 384-well plates (Corning, CLS3985) at $1.6 \times 10^4$ cells per well. Human sera or monoclonal antibodies were serially diluted (2-fold) in DMEM-5% FBS and mixed in duplicate with an equal volume of SARS-CoV-2 virus solution at 2× the median lethal dose ($2 \times LD_{50}$), calculated as indicated above. After 1 h of virus–serum co-incubation at 37 °C, 40 µl was added to an equal volume of pre-plated cells. Cell plates were then incubated for 20 h before direct imaging on an InCell Analyzer HS2500 high-content fluorescence microscopy system (Cytiva). Cellular nuclei counts were obtained with IN Carta automated image analysis software (Cytiva), and the percentage of virus neutralization was calculated with the formula: $\%N = (D - (1 - Q)) \times 100/D$, where 'Q' is a well's nuclei count divided by the average count for uninfected controls (defined as having 100% neutralization) and $D = 1 - Q$ for the average count of positive infection controls (defined as having 0% neutralization). For additional detail and the rationale behind this formula, see Supplementary Fig. 8 in ref. [23]. The cut-off for determining the neutralization end-point titre of diluted serum samples was set to the last consecutive dilution reaching ≥50% neutralization for the average of technical replicates. Unless otherwise specified, samples were tested at a starting dilution of 1/20.

**High-content SARS-CoV-2 microneutralization assay with VeroE6 cells.** Human sera or monoclonal antibodies were serially diluted (2-fold) in MEM-2%FBS and mixed in duplicate with an equal volume of SARS-CoV-2 virus solution at $1.25 \times 10^4$ $TCID_{50}$ ml$^{-1}$. After 1 h of virus–serum co-incubation at 37 °C, 40 µl was added to an equal volume of freshly trypsinized VeroE6 cells pre-seeded in 384-well plates at $5 \times 10^3$ cells per well in MEM-2% FBS (final MOI = 0.05). Cells were then incubated for 72 h and subsequently stained with Hoechst-33342 (Invitrogen, R37605) at a final concentration of 5% v/v. Entire well surface areas were then imaged, nuclei enumerated and % neutralization determined exactly as indicated above for the R-20 assay.

**ALI cultures.** Culture and differentiation of pBEC at the ALI was performed according to previously described methods[30]. Briefly, cells were seeded at $2 \times 10^5$ cells in 12-well plate trans-wells (Corning, 3460) and initially grown submerged in ALI initial media comprising 50% Bronchial Epithelial Basal Medium (BEBM)-50% DMEM containing 0.1% hydrocortisone, 0.1% bovine insulin, 0.1% epinephrine, 0.1% transferrin, 0.4% bovine pituitary extract (all from Lonza SingleQuotes, CC-3171), ethanolamine (final concentration 80 µM), MgCl$_2$ (final concentration 0.3 mM), MgSO$_4$ (final concentration 0.4 mM), bovine serum albumin (final concentration 0.5 mg ml$^{-1}$), amphotericin B (final concentration 250 µg ml$^{-1}$), all-trans retinoic acid (30 ng ml$^{-1}$) and 2% penicillin-streptomycin with 10 ng ml$^{-1}$ recombinant human epithelial growth factor (rhEGF). Upon reaching confluence in the trans-wells (~3–5 d after seeding), the rhEGF concentration was lowered to 0.5 ng µl$^{-1}$ and cells were allowed to differentiate at ALI without media in the compartment. After 25–30 d of differentiation, the quality of culture was confirmed by microscope observation of ciliated epithelium as well as presence of mucus/mucus-producing cells.

**Infection of ALI-pBECs and virus outgrowth assay.** Before infection, cells were washed once with phosphate buffered saline (PBS). Cells were inoculated with SARS-CoV-2 variants at 0.1 MOI, diluted in BEBM minimal media (Lonza; BEBM+1% Insulin-Transferrin-Selenium, 0.5% linoleic acid, 2% penicillin-streptomycin and 1% fungizone) on the apical surface only. After a 2 h incubation at 37 °C, the inoculum was collected and the apical surface was washed with 500 µl PBS to remove unbound virus. pBECs were collected at 3 d and 7 d post infection. Upon collection, apical washes were collected by addition of 500 µl PBS for 5 min. All samples were stored at −80 °C until further use. A virus outgrowth assay was performed in HAT-24 cells to quantify infectious virus in apical washes. Samples were serially diluted 10-fold in culture media (DMEM-5% FBS), with a starting dilution of 1 in 5, and 40 µl of each dilution was transferred to cells pre-seeded in 384-well plates at $5 \times 10^3$ cells per well. Each sample was run in quadruplicate over 8 dilutions. Wells were imaged by brightfield microscopy at 72 h post infection, using an InCell Analyzer HS2500 high-content microscope (Cytiva). Cells were binarily scored by visual examination for CPE (+ or −) and $TCID_{50}$ ml$^{-1}$ values were calculated via the Spearman-Karber method[21].

**Statistical analysis.** Sigmoidal dose-response curves and $IC_{50}$ values were obtained with GraphPad Prism software using the nonlinear regression for inhibitor versus response with variable slope and four parameters. The non-parametric Spearman's coefficient of correlation 'r' was calculated with GraphPad Prism using a two-tailed analysis with a confidence interval of 95%. To compare the immune evasiveness of different viral variants, titre fold reductions were calculated for each participant and variant by dividing the $IC_{50}$ from the participant-matched ancestral virus control by the $IC_{50}$ of each variant. To test for statistical significance, the mean fold-reduction $IC_{50}$s for each variant were compared to that of the 'wildtype' virus control (A.2.2) using a non-parametric Friedman test with Dunn's multiple comparison test. To calculate differences in variant infectivity, PCR Ct values from primary nasopharyngeal swabs were plotted against end-point titres from R-20

assay in HAT-24 cells. Titres were log transformed and linear regression was used to model the relationship between Ct values and titres for each variant. Statistical significance was assessed by comparing the slopes and intercept values for each variant, and linear regression coefficient and Spearman's correlation coefficient were determined (95% confidence interval). Similar analyses parameters were employed to explore the relationship between age and virus titres.

**Estimating vaccine effectiveness against Omicron.** Vaccine effectiveness (VE) against Omicron was estimated using the approach and model previously reported[34,35]. Briefly, we estimated VE for a given vaccine/cohort against a given variant on the basis of the ($\log_{10}$) geometric mean (GM) neutralization titre, normalized to the GM neutralization titre (against ancestral virus) of convalescent individuals (exposed to ancestral virus). This normalized ($\log_{10}$) GM titre is given by $\mu$, and the VE for a given $\mu$ is predicted using the equation

$$VE\,(\mu) = \int_{-\infty}^{\infty} N(x,\,\mu,\,\sigma)\,\frac{1}{1+e^{-k(x-x_{50})}}\,dx,$$

where $N$ is the probability density function of a normal distribution of the distribution in $\log_{10}$ neutralization titres with mean $\mu$ and standard deviation $\sigma$, and $x_{50}$ and $k$ are the parameters of a logistic function representing the $\log_{10}$-normalized neutralization titre associated with 50% protection and $k$ determines the steepness of the logistic function. The parameters $\sigma = 0.46$ (representing the spread of neutralization titres in vaccinated individuals), $x_{50} = \log_{10}(0.20)$ and $k = 3.1$ were estimated previously by fitting vaccine efficacy data from randomized control trials across 7 SARS-CoV-2 vaccines[34]. Thus, vaccine efficacy of BNT162b2-vaccinated individuals against Omicron is estimated by the normalized GM neutralization titre of BNT162b2-vaccinated individuals against Omicron. Previously, we have estimated that BNT162b2-vaccinated individuals have a GM neutralization titre (against ancestral virus) that is 2.4-fold the GM neutralization titre of convalescent individuals (who were exposed to ancestral virus)[34]. Thus, to estimate the VE against Omicron of BNT162b2-vaccinated individuals, we calculated the normalized GM neutralization titre of these individuals against Omicron. A 20.2-fold drop in neutralization against Omicron was estimated to lead to a GM neutralization titre in BNT162b2-vaccinated individuals against Omicron of 2.4/20.2 = 0.119-fold of the GM neutralization titre of convalescent individuals (after exposure to ancestral virus) against ancestral virus (thus, $\mu = \log_{10}(0.119)$ in the above equation to estimate vaccine efficacy for BNT162b2-vaccinated individuals against Omicron). Similarly, we have previously shown that mRNA vaccines in previously infected individuals produced GM neutralization titres that are approximately 12.0-fold higher than the GM titres of convalescent individuals who were exposed to ancestral virus[54]. Thus, the 20.2-fold loss of neutralization to Omicron was estimated to give a GM neutralization titre against Omicron that is 0.594-fold of the GM titre of convalescent participants against ancestral virus (after exposure to ancestral virus). Thus, to estimate vaccine efficacy for previously infected and mRNA-vaccinated individuals, we used $\mu = \log_{10}(0.594)$ in the above equation. Confidence intervals for predicted vaccine efficacies were estimated using bootstrapping.

**Materials availability.** This study generated a new reagent described as HEK293T-ACE2-TMPRSS2-Clone#24 (HAT-24) cell line. HAT-24 cells will be made available through CellBank Australia, ATCC and ECACC. Laboratories unable to source this cell line (or until this cell line is available at these collections) from the latter organizations can directly obtain the cell line through the corresponding author through a standard material transfer agreement.

**Reporting summary.** Further information on research design is available in the Nature Research Reporting Summary linked to this article.

## Data availability

All viral sequences have been deposited with the NIH BioProject under Project #PRJNA824540. GISAID reference numbers are listed in Supplementary Table 10. IC$_{50}$ or cut-off titre values for all cohorts listed herein are in Supplementary Tables 1–9. Source data for figures is available at https://doi.org/10.6084/m9.figshare.19530550.v1. Source data for supplementary data is available at https://doi.org/10.6084/m9.figshare.19523401.v1. Source data are also provided with this paper.

## Code availability

No custom code was generated or used.

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

## Acknowledgements

Viral isolation was in collaboration with NSW Health Pathology, where diagnostics and whole-genome sequencing initially identified clinical material for isolate expansion. In this setting, we collectively acknowledge the contribution of the scientists and pathologists of NSW Health Pathology and the Kirby Institute to this project. Growth and use of primary isolates was under approval of the NSW Chief Health Officer following independent scientific review. This work was primarily supported by Australian Medical Foundation research grants MRF2005760 (S.G.T., M.P.D., G.V.M. and W.D.R.), MRF2001684 (A.D.K. and S.G.T.) and a Medical Research Future Fund Antiviral Development Call grant (D.C. and W.D.R.), Medical Research Future Fund COVID-19 grant (MRFF2001684, A.D.K. and S.G.T.) and the New South Wales Health COVID-19 Research Grants Round 2 (F.B. and S.G.T.). H.-M.J. was supported by grants TRR130 and GRK1660 from the Deutsche Forschungsgemeinschaft (DFG), grants 01KI2043 and COVIM (NaFoUniMedCovid19; FKZ: 01KX2021) from the Bundesministerium für Bildung und Forschung (BMBF), the Kastner Foundation, and grants from the Bayerische Forschungsstiftung and the Bavarian Ministry of Art and Science (CORad).

## Author contributions

Assay development was performed by A. Aggarwal, A.O.S. and S.G.T. Viruses were propagated by A. Aggarwal, A.O.S., G.W., A. Akerman, V. Milogiannakis., S.M., Y.L. and S.G.T. Experiments were performed by A. Aggarwal, A.O.S., G.W., A. Akerman, C.E., V. Milogiannakis., S.M. and S.G.T. Additional research support including sequencing, mAbs and serum were provided by D.L.B., M.R.S., C.S.P.F., F.B., A.P., S.V.H., V. Mathivanan., C.F., A.K., A.C.H., M.L.M., S.A.-C., N.R., G.C., G.P.S., P.S., J.J., H.L., J.Y.H., O.M., H.-M.J., M.P.D., D.R.D., G.V.M., D.S.K., D. Christ., C.C.G., D. Cromer., R.R., D.J.S., N.W.B., W.D.R. and A.D.K. Study analysis was performed by A. Aggarwal, A.O.S., G.W., A. Akerman and S.G.T. The manuscript was drafted by A. Aggarwal, A.O.S., G.W., A. Akerman and S.G.T., with all authors providing editorial support for the published version. The study was supervised by C.C.G., D. Christ., D.J.S., N.W.B., W.D.R., A.D.K. and S.G.T.

## Competing interests

The authors declare no competing interests.

## Additional information

**Correspondence and requests for materials** should be addressed to Stuart G. Turville.

# Reporting Summary

## Statistics

For all statistical analyses, confirm that the following items are present in the figure legend, table legend, main text, or Methods section.

| n/a | Confirmed | |
|---|---|---|
| ☐ | ☒ | The exact sample size (*n*) for each experimental group/condition, given as a discrete number and unit of measurement |
| ☐ | ☒ | A statement on whether measurements were taken from distinct samples or whether the same sample was measured repeatedly |
| ☐ | ☒ | The statistical test(s) used AND whether they are one- or two-sided *Only common tests should be described solely by name; describe more complex techniques in the Methods section.* |
| ☒ | ☐ | A description of all covariates tested |
| ☐ | ☒ | A description of any assumptions or corrections, such as tests of normality and adjustment for multiple comparisons |
| ☐ | ☒ | A full description of the statistical parameters including central tendency (e.g. means) or other basic estimates (e.g. regression coefficient) AND variation (e.g. standard deviation) or associated estimates of uncertainty (e.g. confidence intervals) |
| ☐ | ☒ | For null hypothesis testing, the test statistic (e.g. *F*, *t*, *r*) with confidence intervals, effect sizes, degrees of freedom and *P* value noted *Give P values as exact values whenever suitable.* |
| ☒ | ☐ | For Bayesian analysis, information on the choice of priors and Markov chain Monte Carlo settings |
| ☒ | ☐ | For hierarchical and complex designs, identification of the appropriate level for tests and full reporting of outcomes |
| ☐ | ☒ | Estimates of effect sizes (e.g. Cohen's *d*, Pearson's *r*), indicating how they were calculated |

*Our web collection on statistics for biologists contains articles on many of the points above.*

## Software and code

Policy information about availability of computer code

| Data collection | InCell Analyzer HS2500 (Cytiva) v7.3-17022 |
|---|---|
| Data analysis | GraphPad Prism ver 9.3.0; InCarta (Cytiva) v1.12.3782849 |

For manuscripts utilizing custom algorithms or software that are central to the research but not yet described in published literature, software must be made available to editors and reviewers. We strongly encourage code deposition in a community repository (e.g. GitHub). See the Nature Portfolio guidelines for submitting code & software for further information.

## Data

Policy information about availability of data

All manuscripts must include a data availability statement. This statement should provide the following information, where applicable:

- Accession codes, unique identifiers, or web links for publicly available datasets
- A description of any restrictions on data availability
- For clinical datasets or third party data, please ensure that the statement adheres to our policy

Whole-genome viral sequences for the SARS-CoV-2 variants used in this study have been deposited to GISAID and are publicly available as of the date of publication. Accession numbers are listed in Table S10. IC50 or cut-off titres values for all cohorts listed herein are in Tables S1-9. Source data for figures is available at:  https://doi.org/10.6084/m9.figshare.19530550.v1Source data for supplementary data is available at: https://doi.org/10.6084/m9.figshare.19523401.v1

# Field-specific reporting

Please select the one below that is the best fit for your research. If you are not sure, read the appropriate sections before making your selection.

☒ Life sciences ☐ Behavioural & social sciences ☐ Ecological, evolutionary & environmental sciences

For a reference copy of the document with all sections, see nature.com/documents/nr-reporting-summary-flat.pdf

# Life sciences study design

All studies must disclose on these points even when the disclosure is negative.

| | |
|---|---|
| Sample size | Of the >200 ADAPT participants, a panel of 25 representative donors was chosen based on serum neutralization titres against 'widtype' B.1 (B.1.319)-clade virus (4 donors each for endpoint titres of: ≤40, 40, 80, 160, 320 and ≥640). A second cohort was composed of 24 healthy adult vaccine-recipients, who received the BNT162b2 vaccine in 2021. Serum samples were collected three weeks post second-dose vaccination. In the convalescent cohort (from ADAPT) only donors that showed positive serum neutralization titres against 'widtype' B.1-clade virus (B.1.319) were used for calculating fold changes in IC50 for variants relative to the 'wild type' (n=15). All donors in this second cohort were included for analysis as they all had positive serum neutralization titres against 'widtype' B.1-clade virus (B.1.319). In cohort 1 and in cohort 2, case sample size was determined to be sufficient for statistical testing. Five IVIG lots (Poly IgG 1033, 4850, 7450, 0301, 0723) manufactured using the Privigen process included US plasma collected by plasmapheresis from a mixture of vaccinated with SARS-CoV-2 mRNA vaccines, convalescent and non-convalescent donors (source plasma, n between 9495-23,667 per batch) with majority of donations collected between April and June 2021. The sample size for nasopharyngeal swabs was dictated by availability of swab specimens from community donors. For the Avalon cluster swabs from 17 donors were tested while for the Delta cluster 80 swab specimens were collected and tested. For Omicron studies, 35 nasopharyngeal swab samples were analysed in this study. |
| Data exclusions | In the convalescent cohort (ADAPT) 10 donors that did not reach titre with any of the variants except the ancestral strain were excluded while calculating fold changes in neutralization titres (Figure 4 in the manuscript) as it was not possible to calculate IC50 values for these samples. |
| Replication | Each experiment was run at least thrice and samples were run in quadruplicates. All repeat experiments were successful as observed with appropriate positive and negative controls. |
| Randomization | Samples from the ADAPT cohort were selected on the basis of serum neutralization titres against 'widtype' B.1-clade virus. As the samples were primarily used for assay cross-validation, it was important to determine neutralisation titers that covered a range of potency. For other samples, they were allocated randomly (vaccine samples and postive nasopharygeal swabs). |
| Blinding | Investigators were blinded to samples throughout. The only data available for the samples was within the ADAPT cohort, where neutralisation titers were determined from a larger donor pool. From this pool a continuum of responses was assembled as outlined above. |

# Reporting for specific materials, systems and methods

We require information from authors about some types of materials, experimental systems and methods used in many studies. Here, indicate whether each material, system or method listed is relevant to your study. If you are not sure if a list item applies to your research, read the appropriate section before selecting a response.

## Materials & experimental systems

| n/a | Involved in the study |
|---|---|
| ☐ | ☒ Antibodies |
| ☐ | ☒ Eukaryotic cell lines |
| ☒ | ☐ Palaeontology and archaeology |
| ☒ | ☐ Animals and other organisms |
| ☐ | ☒ Human research participants |
| ☒ | ☐ Clinical data |
| ☒ | ☐ Dual use research of concern |

## Methods

| n/a | Involved in the study |
|---|---|
| ☒ | ☐ ChIP-seq |
| ☒ | ☐ Flow cytometry |
| ☒ | ☐ MRI-based neuroimaging |

# Antibodies

| | |
|---|---|
| Antibodies used | AB-3467, Sotrovimab, Casirivimab, Imdevimab, Bamlanivimab, Cilgavimab and Tixagevimab |
| Validation | AB-3467 has been extensively characterised in Burnett et al ( DOI: 10.1016/j.immuni.2021.10.019 ). Sotrovimab, Casirivimab, Imdevimab, Bamlanivimab, Cilgavimab and Tixagevimab were generated by gene synthesis, cloned into human IgG1 expression vectors, and produced in ExpiCHO cells. After production in ExpiCHO cells, monoclonal antibodies were characterized for binding to recombinant RBD by biolayer-interferometry (BLI) and for neutralization of live early clade (A2.2) SARS-CoV-2 virus |

# Eukaryotic cell lines

Policy information about cell lines

| | |
|---|---|
| Cell line source(s) | Hek 293T cells (Thermofisher Scientific, #R70007)<br>VeroE6-TMPRSS2 (CellBank Australia, JCRB1819)<br>VeroE6 cells (ATCC® CRL-1586™)<br>Primary bronchial epithelial cells (pBEC) from P. A. B. Wark (University of Newcastle)<br>ExpiCHO cells (Thermofisher, A29133) |
| Authentication | The Garvan Molecular Genetics facility at the Garvan Institute of Medical Research performed cell line authentication on all human cell lines used. DNA from each cell line was analysed for short tandem repeat loci using the PowerPlexR 18D System. All human cell lines listed above were >80% identical, indicating they originated from the cell line specified. |
| Mycoplasma contamination | All cell lines were tested at the Mycoplasma Testing Facility at UNSW using the Lonza MycoAlertTM Mycoplasma Detection Kit (catalogue number LT07-318). All cell lines used in this study were negative for mycoplasma contamination. |
| Commonly misidentified lines (See ICLAC register) | No misidentified cell lines were used in this study. |

# Human research participants

Policy information about studies involving human research participants

| | |
|---|---|
| Population characteristics | For ADAPT cohort, this has been described in detail in Tea et al ( DOI: 10.1371/journal.pmed.1003656 ). For the samples used in each setting, we have stated median age, interquartile age range and also the female to male ratio. This is outlined in the supplementary tables.<br>Vaccinated donors were drawn from laboratory and healthcare workers, as they were the first groups in Australia to be vaccinated. For the samples used in each setting, we have stated median age, interquartile age range and also the female to male ratio. This is outlined in the supplementary tables.<br>US plasma donor numbers in Polyclonal IgG batches are outlined in the supplementary figures. The median age and female:male ratio is not known. |
| Recruitment | Convalescent patients from ADAPT were diagnosed at a community-based fever clinic and recruited as outlined by Tea et al ( DOI: 10.1371/journal.pmed.1003656 ). The donors tested herein were assigned based on initial neutralisation titers to the B.1-clade (B.1.319) virus. Donors with high neutralisation titers to early clade variants are biased to being male with a median age of 57. Vaccine donors derived from laboratory and healthcare workers are approximately 2:1 female to male in ratio with a median age of 38. |
| Ethics oversight | All human serum samples were obtained with written informed consent from the participants and was approved by the St Vincent's Hospital Ethics Committee (2020/ETH00964; 2018/ETH00145; 2021/ETH00180).  The use of remnant diagnostic swabs for assessment and development of diagnostic tests, for determining viral genotype, and for in vitro and in vivo research assessing phenotypes such as its determinants of transmissibility, pathogenicity and response to preventive and treatment interventions was under the approval of the NSW CHO following independent scientific review 2021/NSWCHO H21/126831 ). |

Note that full information on the approval of the study protocol must also be provided in the manuscript.

