## [Peer Review File · Nature Microbiology]

Peer Review Information

Journal: Nature Microbiology

Manuscript Title: An accessible high content platform for rapid isolation and characterisation of immune evasion and fitness of SARS-CoV-2 variants

Corresponding author name(s): Professor Stuart Turville

Reviewer Comments & Decisions:

Decision Letter, initial version:
--

18th February 2022

Dear Stuart,

Thank you for your patience while your manuscript "Rapid isolation and resolution immune evasion and viral fitness across contemporary SARS-CoV-2 variants" was under peer-review at Nature Microbiology and my apologies for the delay in the review process while we were waiting for the

reviewers' comments. It has now been seen by 3 referees, whose expertise and comments you will find at the of this email. You will see from their comments below that while they find your work of interest, some important points are raised. We are very interested in the possibility of publishing your study in Nature Microbiology, but would like to consider your response to these concerns in the form of a revised manuscript before we make a final decision on publication.

In particular, you will see that reviewer #2 raises a concern about the applicability/availability of this approach for other public health labs that we will need to be carefully addressed. The rest referees' reports are clear and the remaining issues should be straightforward to address. As discussed, we would like to ask you to please include your related (and revised) submission (NMICROBIOL-21123040B-Z) into the revised version of this submission and to also include the point-by-point response of the related submission. Please send the merged and revised manuscript back to us as fast as possible as I'm sure you are aware this is a very time-sensitive topic. Please let me know if you expect any delays or have additional questions.

If you have not done so already please begin to revise your manuscript so that it conforms to our Article format instructions at <http://www.nature.com/nmicrobiol/info/final-submission/>

The usual length limit for a Nature Microbiology Article is six display items (figures or tables) and 3,000 words. We have some flexibility, and can allow a revised manuscript at 3,500 words, but please consider this a firm upper limit. There is a trade-off of ~250 words per display item, so if you need more space, you could move a Figure or Table to Supplementary Information.

Some reduction could be achieved by focusing any introductory material and moving it to the start of your opening 'bold' paragraph, whose function is to outline the background to your work, describe in a sentence your new observations, and explain your main conclusions. The discussion should also be limited. Methods should be described in a separate section following the discussion, we do not place a word limit on Methods.

Nature Microbiology titles should give a sense of the main new findings of a manuscript, and should not contain punctuation. Please keep in mind that we strongly discourage active verbs in titles, and that they should ideally fit within 90 characters each (including spaces).

Please include a data availability statement as a separate section after Methods but before references, under the heading "Data Availability". This section should inform readers about the availability of the data used to support the conclusions of your study. This information includes accession codes to public

repositories (data banks for protein, DNA or RNA sequences, microarray, proteomics data etc...), references to source data published alongside the paper, unique identifiers such as URLs to data repository entries, or data set DOIs, and any other statement about data availability. At a minimum, you should include the following statement: "The data that support the findings of this study are available from the corresponding author upon request", mentioning any restrictions on availability. If DOIs are provided, we also strongly encourage including these in the Reference list (authors, title, publisher (repository name), identifier, year). For more guidance on how to write this section please see:

<http://www.nature.com/authors/policies/data/data-availability-statements-data-citations.pdf>

To improve the accessibility of your paper to readers from other research areas, please pay particular attention to the wording of the paper's opening bold paragraph, which serves both as an introduction and as a brief, non-technical summary in about 150 words. If, however, you require one or two extra sentences to explain your work clearly, please include them even if the paragraph is over-length as a result. The opening paragraph should not contain references. Because scientists from other sub-disciplines will be interested in your results and their implications, it is important to explain essential but specialised terms concisely. We suggest you show your summary paragraph to colleagues in other fields to uncover any problematic concepts.

If your paper is accepted for publication, we will edit your display items electronically so they conform to our house style and will reproduce clearly in print. If necessary, we will re-size figures to fit single or double column width. If your figures contain several parts, the parts should form a neat rectangle when assembled. Choosing the right electronic format at this stage will speed up the processing of your paper and give the best possible results in print. We would like the figures to be supplied as vector files - EPS, PDF, AI or postscript (PS) file formats (not raster or bitmap files), preferably generated with vector-graphics software (Adobe Illustrator for example). Please try to ensure that all figures are non-flattened and fully editable. All images should be at least 300 dpi resolution (when figures are scaled to approximately the size that they are to be printed at) and in RGB colour format. Please do not submit Jpeg or flattened TIFF files. Please see also 'Guidelines for Electronic Submission of Figures' at the end of this letter for further detail.

Figure legends must provide a brief description of the figure and the symbols used, within 350 words, including definitions of any error bars employed in the figures.

When submitting the revised version of your manuscript, please pay close attention to our [href="https://www.nature.com/nature-research/editorial-policies/image-integrity">Digital Image Integrity Guidelines](https://www.nature.com/nature-research/editorial-policies/image-integrity) and to the following points below:

Please include a statement before the acknowledgements naming the author to whom correspondence and requests for materials should be addressed.

Finally, we require authors to include a statement of their individual contributions to the paper -- such as experimental work, project planning, data analysis, etc. -- immediately after the acknowledgements. The statement should be short, and refer to authors by their initials. For details please see the Authorship section of our joint Editorial policies at http://www.nature.com/authors/editorial_policies/authorship.html

- * include a point-by-point response to any editorial suggestions and to our referees. Please include your response to the editorial suggestions in your cover letter, and please upload your response to the referees as a separate document.

- * ensure it complies with our format requirements for Letters as set out in our guide to authors at www.nature.com/nmicrobiol/info/gta/

- * state in a cover note the length of the text, methods and legends; the number of references; number and estimated final size of figures and tables

- * resubmit electronically if possible using the link below to access your home page:

[Redacted]

- * This url links to your confidential homepage and associated information about manuscripts you may have submitted or be reviewing for us. If you wish to forward this e-mail to co-authors, please delete this link to your homepage first.

Please ensure that all correspondence is marked with your Nature Microbiology reference number in the subject line.

Nature Microbiology is committed to improving transparency in authorship. As part of our efforts in this direction, we are now requesting that all authors identified as 'corresponding author' on published papers create and link their Open Researcher and Contributor Identifier (ORCID) with their account on the Manuscript Tracking System (MTS), prior to acceptance. This applies to primary research papers only. ORCID helps the scientific community achieve unambiguous attribution of all scholarly contributions. You can create and link your ORCID from the home page of the MTS by clicking on 'Modify my Springer Nature account'. For more information please visit www.springernature.com/orcid.

We hope to receive your revised paper within three weeks. If you cannot send it within this time, please let us know.

Yours sincerely,

[Redacted]

Reviewer Expertise:

Referee #1: vaccines

Referee #2: coronaviruses, serologic testing

Referee #3: vaccines, serologic testing

Reviewers Comments:

Reviewer #1 (Remarks to the Author):

The authors present a research article entitled "Rapid isolation and resolution of immune evasion and viral fitness across contemporary SARS-CoV-2 variants". In this study, the authors have introduced a rapid platform (R-20) for rapid SARS-CoV-2 isolation even at low viral load and characterization of all variants of concern (VOC) including six variants of interest (VUI). The main strength of this platform is greater sensitivity to viral induced cytopathic effect (CPE) of HAT24 clone compared to traditional VeroE6 cell line. Overall, the observations found from immune evasion of all major VOCs across both vaccinated and convalescent donors are quite interesting which will further inform vaccine efforts. We suggest to include a specific "limitations section" to the manuscript that reflects the limitations of this study and also to add ethical statement of this study at the beginning of the method section.

Comments for the Authors

1. Both HAT24 and HAT10 are HEK293T clones co-expressing ACE-2 and TMPRSS2. What is the reason behind hyper-permissive and sensitive to SARS-CoV-2 infection of HAT24 clone compared to HAT10 clone? Moreover, the authors mentioned CPE within 8 hours post infection for HAT24 clone in supplementary figure 1. So why has the platform been designed considering only 20 hours of post infection?
2. The authors mentioned that HAT24 clone will be able to isolate SARS-CoV-2 from 80% of PCR positive swabs. What are Ct values of these 20% PCR positive swabs? Is there any relation of virus isolation with high Ct value?
3. In this study, authors reported expansion of B.1.517.1 variant in December 2020 and in July 2021 due to community outbreak in Australia and also calculated Delta to be 9.1 fold more infectious virus than the B.1.517.1. A look at the neutralization antibody response of B.1.517.1 variant among vaccinated and convalescent donors needs to be added? It will be great if this would also include neutralization antibody response of this variant in figure number 4.

Overall the novel R20 platform introduced in this study is promising and will be used widely for rapid isolation of future outbreaks to minimize the global spread of SARS-CoV-2.

Reviewer #2 (Remarks to the Author):

Aggarwal and colleagues describe the use of a genetically manipulated cell lines (HAT-24) as a platform for isolating/characterizing SARS-CoV-2 variants. This cell line is highly sensitive to SARS-CoV-2 infection, which allows having readouts from their assays with 24 hours (R-20). Thus, this platform might have a great potential to provide some timely and important results for newly emerge SARS-CoV-2 variants.

This reviewer, however, has some major concerns:

1. The authors actually did not know why HAT-24 is so susceptible to SARS-CoV-2 infection. For this kind of public health studies, it is ok. But the authors should briefly mentioned this in the discussion. In addition, it is not right to say that this cell line "lacked viral restrictions" (Result section). There would be many possible hypotheses to explain this (e.g. increased susceptibility for infection and increase susceptibility to commit cell death).
2. This cell line is unique. As the mechanism of action is unknown, it is impossible for other to generate similar cell lines for the purpose. How can other public health labs can be benefit from this work?
3. The authors use Vero-E6 as a control cell line as a control. This cell line is well known to be not efficient for virus isolation etc. The author actually compared Vero-E6, Vero-E6-TMPRSS2 and HAT-24 side-by-side (Section "Rapid resolution of in vivo....."). It would be more useful if the authors can show the data and show them in the first part of their result section.
4. Any systematic analysis of the viral replication kinetic in HAT-24 (with high and low MOI)?
5. Did the authors sequence any virus isolated from this platform? Are these sequences still identical to those from the original clinical samples?

Minor concerns:

1. The Y axis of Fig 2D and 2E should have the range of scale.
2. It is bias to cite this preprint (<https://www.medrxiv.org/content/10.1101/2021.09.07.21263229v1.full.pdf>) . There are many studies show that VeroE6-TMPRSS2 can manage to isolate virus from samples with a higher CT value.
3. Figure 6K should be Figure 5K.
4. Figures 1A, 6A and 6B are not necessary and they can be deleted.

5. Figure 6F. What was the time for these cultures, 20 or 96 hrs?

Reviewer #3 (Remarks to the Author):

Summary: The authors have developed a HEK293T-based cell line for the rapid, efficient rescue of SARS-CoV-2 viral isolates and demonstrated its utility using a large cohort of quarantine surveillance nasopharyngeal swab samples. This cell line facilitated rapid isolation of SARS-CoV-2 isolates including all relevant variants of concern and interest, from even low titer samples. The authors demonstrated this rapid short term culture platform provided viral end point titer data that correlated well with standard Vero cultures. The platform was used to examine immune evasion of VOCs from a panel convalescent and vaccinee donor sera and further could be used to assess viral infectivity. Using these newly developed approaches, the authors were able to quickly isolate and characterize the Omicron variant as it emerged, demonstrating it to be the most evasive variant to antibody neutralization and confirming Omicron's shift in the mode of fusion/cell entry.

Critique: The study is a very well designed and elegant approach that not only provided a valuable tool for rapid community surveillance, but also was used to generate a very nicely controlled data set for examining viral phenotypes. I have no major critiques, but only minor points for correcting some confusion and perhaps allowing for better communication of certain aspects of the data.

Minor points:

1. Are there other rescue cell lines for SARS-CoV-2 beyond the standard Vero cultures that have been examined/developed? I am not aware of the entirety of this direction of research and it may be worth a couple of sentences in the intro to address this and cite relevant comparators.
2. I found the implication that the R-20 platform could accurately assess fitness to be confusing. The authors point out that the rapid rescue with a hyperpermissive cell line has restricted the 'rounds of infection' as compared to the standard Vero cell line. Additionally, as shown with Omicron, the mode of entry can also impact this variable. I may be missing something in how this is communicated, but it seems to me that viral fitness is too multivariable to assess here. The infectivity phenotype data, using PCR Ct values as a readout for comparison is strong, but fitness assessment or the implication that it can be accurately assessed does not seem to be supported or at minimum, communicated well.
3. Figure 4 labels appear to be wrong. Panels G-J state B.1.319 as one of the VOCs examined whereas B.1.2 is listed in A-D. These appear inconsistent. Also, in the main text and in the figure legend, it is not immediately clear that the "fold reduction" is comparing. I assume as compared to A2.2, but the text is not clear in this section.
4. Main text states that "across all vaccine recipients, breadth was similar across all donors tested". I am not entirely clear on what this means. I do see some consistencies but the text should be clear about similarities and differences that are being pointed out here.

Author Rebuttal to Initial comments

Dear [Redacted],

Re. NMICROBIOL-21123040B-Z & NMICROBIOL-21123138

Thank you for the time as the handling editor for our Manuscripts entitled “SARS-CoV-2 Omicron: evasion of potent humoral responses and resistance to clinical immunotherapeutics relative to viral variants of concern” (NMICROBIOL-21123040B-Z) & Rapid isolation and resolution immune evasion and viral fitness across contemporary SARS-CoV-2 variants (NMICROBIOL-21123138).

To summarise the two manuscripts briefly, *NMICROBIOL-21123138* was a platform that we started developing in late 2020. This was used and validated extensively in 2021 to help the Covid effort in 2021 at several levels. Firstly, quarantine surveillance of variants and their relative risk. Secondly to facilitate screening of Australian wide cohorts that were potentially vulnerable due to their low vaccine responses (cancer treatment and other states where the immune system is suppressed). Thirdly, rapidly validate and test existing and future clinical immunotherapeutics. This included monoclonal base therapeutics and those developed from thousands of vaccine donors.

In contrast *NMICROBIOL-21123040B-Z* was using the platform to rapidly characterise Omicron. This was in late November when Omicron was initially observed spreading worldwide and there were no observations. The latter was submitted as a Letter and we uploaded this as a pre-print on the 14th of December. As I will outline further below, this was at a time where only a few laboratories had and were sharing their data on Omicron. Based on the Covid response, our reports and others independently verified observations on Omicron and this was then embedded in both national and international guidelines for vaccination and therapeutics (e.g. CDC guidelines).

During the review process and in communication with you and your editorial colleagues, we collectively decided to merge the two papers and in doing so map out the platform and then a snapshot of all of the major variants circulating in 2021, which of course included Omicron.

In this rebuttal, you have requested we address both reviews. This was two reviews of *NMICROBIOL-21123040B-Z* and three reviews of *NMICROBIOL-21123138*. To avoid confusion, we will address each as separate rebuttals and how we have addressed each point in the revised and merged manuscript.

A. *Rebuttal for NMICROBIOL-21123040B-Z “SARS-CoV-2 Omicron: evasion of potent humoral responses and resistance to clinical immunotherapeutics relative to viral variants of concern”.*

Whilst the majority of this reply is from the team who did the vaccine efficacy calculations, I wish to initially address the comment raised by reviewer 2:

1- *“this neutralisation data does not differ from the enormous wealth of knowledge regarding the extreme neutralisation resistance of Omicron. It is for this reason that I am skeptical about the suitability of this manuscript for Nature Microbiology.”*

Response: This work was done primarily at a time when no laboratory worldwide had reported any neutralisation data on Omicron. Our first results came in early December following receipt of a nasopharyngeal swab of the first Australian patient detected on a flight from Doha on the 29th of November. Using the rapid platform we had developed, we observed our first results within one week and had tested our cohort of 78 donor samples, 5 clinical hyperimmune IgG preparations and 6 clinically used monoclonal antibodies.

Our results were rapidly reported to state and national chief health officers and our work was submitted to a pre-print server finally on the 14th of December. This work was followed closely by Australian national media as we were alongside the first laboratories worldwide to report on Omicron (<https://www.smh.com.au/national/nsw/inside-sydney-s-ultra-secure-lab-where-scientists-put-omicron-through-its-paces-20211210-p59gim.html>).

Globally our work, alongside only a few global laboratories were cited within CDC guidelines on the 24th of December, WHO technical reports on the 14th of January, IATA, Stanford Coronavirus Antiviral and resistance database, and many national guidelines regarding vaccination and the appropriate use of therapeutics. In Australia alone our work has been used to develop not only national vaccination but also therapeutic guidelines. Of note Max Kozlov wrote in Nature a News commentary on the 21st of December and interviewed both myself and Olivier Schwartz (senior author of the Planas et al work). This highlights clearly our observations were only amongst a handful and this submission has already had significant impact and also guided us (along with early work at the same time) in the pandemic response.

Whilst I can understand that the reviewer's comments of our Omicron letter that is being reviewed formally in January/February, this needs to be seen in the context that we were going through a formal review process rather than one expedited over only a few days. In contrast, if we take Olivier Schwartz's team's work (Planas et al) that was uploaded as a preprint on the same day as our work but went through an expedited review in Nature (our work is fortunately cited by his team though). Also, if we take the work above cited as the seminal contributions to Omicron by the first reviewer (Carreño et al. Nature, Schmidt et al. NEJM, Cameroni et al. Nature), they appeared as preprints two days before (Schmidt et al), 1 day before (Cameroni et al) or six days after (Carreno et al) our pre-print.

We will now outline point by point discussion of the vaccine efficacy and other points raised by both reviewers.

Reviewers Comments:

Reviewer #1 (Remarks to the Author):

In this manuscript by Aggarwal et al., the authors sought to determine the relative neutralizing capabilities of antibody responses after different variant exposures via infection or after vaccination. The neutralizing antibody titers are highly consistent with other studies (Carreño et al. Nature, Schmidt et al. NEJM, Cameroni et al. Nature) and provide some new insight into antibody titers based on different waves. The title of the manuscript and the abstract imply this is all the study is about, but the authors delve into making vaccine efficacy (should be labeled effectiveness) estimates. This section of the paper became very confusing as there are very few data points that the authors are using to make these estimates. How the authors made these estimates is also not explicitly clear, so it is hard to discern how accurate these estimates are. Of course, other studies (for example Thompson et al. MMWR 2022) have also shown VE against omicron in these settings as well, albeit mostly after the submission of this manuscript. Lastly, the authors have lost sight of what protective immunity is and imply that loss of neutralizing antibodies is equal to no protection, due to the lack of analysis of groups that did not have neutralizing antibodies.

Major Concerns

1. It is unclear how many donors per group were collected and tested. Figure 4 suggests that it was only a handful per group. This should be clearly written in the manuscript and would be very useful to add to Figure 3 (not just the figure legend).

Response: We agree that in this letter immediate clarity on the donor numbers would strengthen this section here and this can be easily done in a revised manuscript. In brief, in late November of 2021, we had completed screening the ADAPT convalescent cohort in Sydney, which consisted of over 200 convalescent patients that have been followed longitudinally up to vaccination from three temporally separate waves of the pandemic which had three distinct viral variants. Each group we assembled as a set of 10 donors from each distinct variant wave. In addition, we also assembled two subsets of convalescent donors (pandemic wave 1 and 2 in Australia) that had also now been vaccinated. This was also compared with 28 non-convalescent health care workers after their third Pfizer vaccine dose.

All were peak humoral responses of patients with high clinical resolution and variant linkage or at the peak response following their last vaccine dose. This was also coupled with hyper-immune globulins from plasma donors within the US. The latter was a very powerful humoral snapshot, as there were batches that were the sum of greater than 20,000 donors.

To ensure immediate clarity of donor numbers we have included n values in each sub-panel in the new merged manuscript. This now appears as Fig. 7.

2. It is unclear where the patient data for VE is coming from. The text reads that it is the just those donors in this study, but there are very few donors. In that case, what proportion would have had a symptomatic infection or severe infection? It likely would be very low if is only a few donors. VEs are usually based on 100s, if not 1000s of data points. Therefore, these estimates seem to be wildly speculative.

Response: We do realise the first reviewer had commented primarily on the vaccine efficacy calculations. That said, removing this section does not weaken from the primary biological observations of this submission. Rather this section

was simply another set of analysis from an analytics team that have developed very powerful analysis tools that can be used in predicting efficacy. Of note, our study predicted vaccine effectiveness reductions of Omicron to 37.2%. Shortly after, early epidemiological data in South Africa supported a drop in real-world vaccine efficacy of 33%. For our biological observations, this analysis was a powerful re-enforcement that our results were within range of epidemiological data. To add, there has been much discussion of teams overestimating fold reductions of variants. Indeed, the first reports from South Africa reported a greater than 40 fold drop in neutralisation towards Omicron, whereas ourselves and others observed fold reductions of around 20. So by using this analytics approach, we had the confidence that what fold reductions we were observing were consistent with real epidemiological data. Below are specific comments regarding their analysis.

Reviewer 1's major concerns here and below relate to the method we used to estimate vaccine efficacy. We did not run a breakthrough infection cohort study estimating vaccine effectiveness and correlating with neutralisation titers, rather as has proved effective for other variants, we used an immunobridging approach to predict vaccine efficacy directly from the drop in neutralising antibodies. As described in the manuscript: "Using this value [estimated drop in neutralisation titre], and the efficacy curve in Khoury et al (Khoury et al, Nature Medicine 2021) we estimated the efficacy and confidence intervals for BNT162b-vaccinated or boosted individuals (in the first few months after vaccination) (Table 1)."

The model we have used to predict efficacy (Khoury et al, Nature Medicine 2021), has been Cited in multiple WHO / CDC / FDA documents on correlates of immunity and predicting protection from COVID-19. More recently we have validated that if we can measure and adjust for the 'fold drop' in neutralisation titres to the variants, the model is predictive of vaccine efficacy against pre-Omicron VOC (Cromer et al, Lancet Microbe 2021). In the present work, we find a mean fold drop in neutralisation of 20.2-fold between the ancestral and Omicron variants. Therefore, we used this fold-drop in the validated model to predict vaccine efficacy against the Omicron variant (as shown in Table 1).

The computational approach was described in detail in the methods, and versions of the model are freely available on Github.

In the revised merged manuscript, we have clarified modelling predictions and stated vaccine effectiveness to make it clear that these were estimated from the drop in neutralisation titre, and not directly measured in our cohort and we cite the relevant published articles that outline this modelling in detail.

"We have previously demonstrated that the neutralising antibody titre is highly predictive of protection from symptomatic SARS-CoV-2 infection¹, and that the drop in titre to a given variant can be used to predict protection against the variant². Therefore, we used this model to predict the loss of vaccine efficacy against Omicron due to the 17.3-fold drop in neutralising antibody titre observed here and incorporating the confidence intervals in this estimate. Using the protection curve in Khoury et al¹ we estimated the level of protection for BNT162b2-vaccinated or boosted individuals in the first few months after vaccination (Table II). This predicts a substantial drop in protection from symptomatic infection for BNT-162b2 vaccinated and boosted individuals (compared to ancestral virus)³, although a relative preservation of protection from severe infection."

3. *It's curious that the CIs for the VE were calculated using a bootstrap analysis, which implies uneven sampling and/or very small numbers in a given group. Again, clear indication of the number of data points will improve the clarity on these estimates.*

Response: The numbers for each group are outlined above. The reason for bootstrapping is that confidence intervals on some non-linear models are extremely hard to determine analytically. Thus we used a bootstrapping approach on the data (as we used previously in both Khoury et al, Nature Medicine 2021, and Cromer et al, Lancet Microbe 2021) to estimate the confidence intervals.

4. *It is also misleading to not look at VE of other groups despite their lack of neutralizing antibodies. Neutralizing antibodies are the most effective means of protection but non-neutralizing antibodies and T cells still play a critical role in limiting symptomatic infections and severe disease.*

Response: We thank the reviewer for pointing out this area of confusion in the field. There is an unfortunate misconception that the inability to measure neutralising antibodies in a given *in vitro* assay implies there is no neutralisation present and a 'lack of protection'. However *in vitro* live virus neutralisation assays can only detect serum neutralisation at the lowest dilution that is used (Here it is 1:20). This is simply because using higher concentrations of serum affects the live cell assay. This is not unique to the platform described herein, but applicable to all live cell assays. For example, spike binding or ACE-2 inhibition is can be detected below the limit of detection of a neutralisation assay (and there is no 'quantum loss' of binding at the detection limit). Importantly, for most assays the threshold for 50% protection from symptomatic SARS-CoV-2 infection actually lies below the detection limit of the assay (Khoury et al Nat Med 2021). Thus, the lack of detection does not imply the lack of neutralising antibodies (or protection).

Importantly, the relationship between neutralisation and protection allows us to predict efficacy below the detection threshold in Omicron neutralisation. The recent empirical studies of vaccine protection in these groups demonstrates this approach is correct. In the revised manuscript we have clarified that a lack of detection of neutralisation *in vitro* does not imply there is no neutralisation capacity of serum (just that it is below the LOD of the assay) as follows.

Minor Concerns

1. *Figures 1 and 2 generally seem out of place. These are proof of principle experiments that could be included in supplemental.*

Response: As this letter is now merged in a more comprehensive study of all variants that were circulating in 2021, the figures are either removed (Fig. 1) or have been moved to the supplement of the new submission (Fig. 2).

2. *The authors are attempting to measure vaccine effectiveness, not vaccine efficacy. Efficacy is only attainable in highly controlled clinical trials.*

Response: Thank you. We note that the model used to estimate protection (Khoury et al Nat Med 2021) was parameterised from seven RCT of different vaccines (which directly estimated efficacy). In addition, data from an observational study of protection in convalescent individuals was also used. Thus, the model was parameterised to predict efficacy (derived from RCT, with the exception of the convalescent individuals). In a revised manuscript we have changed the text throughout from 'efficacy' to 'effectiveness' to reflect the reviewer's comment.

3. *Lack of discussion on what protective immunity against SARS-CoV-2 is limits the impact of this study. Neutralizing antibodies are only one part of this.*

Response: we agree with the reviewer. At the time of submission we were simply reporting on neutralising antibodies. In the merged manuscript the discussion has significantly changed and during review of this letter and the larger manuscript it is apparent that protective immunity is multi-factorial and fortunately there is now a significant body of work that supports this in break-through Omicron infections globally. We have concisely referred to this in the discussion as follows:

“Whilst Omicron represented a significant challenge to the existing two dose vaccination strategy, it fortunately has not translated to the mortality observed in unvaccinated communities in the early phases of the pandemic ⁴. Whilst infection in vaccination populations is consistent with Omicron’s ability to evade humoral immunity, disease severity following infection is multifactorial and would be tempered by both humoral and cellular immunity acquired through vaccination and/or prior infection. . “

Reviewer #2 (Remarks to the Author):

SARS-CoV-2 Omicron: evasion of potent humoral responses and resistance to clinical immunotherapeutics relative to viral variants of concern

This manuscript describes development of a rapid SARS-CoV-2 virus isolation method as well as the live virus neutralisation of multiple SARS-CoV-2 variants, including the now globally dominant Omicron variant of concern. The study was extensive, in that serological specimens from individuals with matched SARS-CoV-2 sequence data were used. Samples from a community-based longitudinal cohort, which included sampling after convalescence, samples from vaccinated health care workers and polyclonal hyperimmune IgG were used to characterise neutralisation sensitivity.

While the manuscript is well written and the figures are clear, and although the developed of an engineered 293T cell for isolation of SARS-CoV-2 is interesting, this neutralisation data does not differ from the enormous wealth of knowledge regarding the extreme neutralisation resistance of Omicron. It is for this reason that I am skeptical about the suitability of this manuscript for Nature Microbiology.

Minor comments:

1. *Introduction Line 69 – would be better to refer to the announcement of the variant – 25 November 2021 rather than date first identified, as this is constantly changing due to retrospective sequencing. For example, currently first date of Omicron detection is from the 1 Nov in the UK.*

In the merged revised manuscript we simply refer to the first positive case detected in Australia that we had access to. We agree that retrospective sequencing of many samples will change how we understand how and when Omicron started to spread globally.

2. *Line 76-85 – should mention the extensive changes in the NTD of Omicron, which also mediate immune escape.*

Response: Yes, the NTD of omicron has been reported to mediate immune escape, however we have focussed more on the site of the Class III antibody Sotrovimab as it was the only antibody that retained significant neutralisation capacity.

3. *Remove Figure 3 references from the introduction (line 100, 102, 105) or make these figure 1 – typically figures are referred to in the order that they are presented in a manuscript.*

Response: As the manuscript has been merged, figure 3 and its appearance now no longer appear.

4. *Line 276 – The authors argue that boosting is needed (potentially with a variant-specific dose). I would argue that boosting is not required in the general population, because with each subsequent variant, we have seen that prevention of infection is not practical as maintenance of high Ab titres are needed to mediate this. Rather the authors should advise that while neutralising humoral antibody titres are reduced against Omicron, the maintenance of binding Ab and T cell epitopes may be linked to the decoupling of cases and hospitalisations/deaths that has been seen globally.*

Response: this discussion no longer appears in the merged manuscript.

B. *Rebuttal for Rapid isolation and resolution immune evasion and viral fitness across contemporary SARS-CoV-2 variants (NMICROBIOL-21123138).*

Reviewer 1. (Remarks to the Author):

The authors present a research article entitled “Rapid isolation and resolution of immune evasion and viral fitness across contemporary SARS-CoV-2 variants”. In this study, the authors have introduced a rapid platform (R-20) for rapid SARS-CoV-2 isolation even at low viral load and characterization of all variants of concern (VOC) including six variants of interest (VUI). The main strength of this platform is greater sensitivity to viral induced cytopathic effect (CPE) of HAT24 clone compared to traditional VeroE6 cell line. Overall, the observations found from immune evasion of all major VOCs across both vaccinated and convalescent donors are quite interesting which will further inform vaccine efforts. We suggest to include a specific “limitations section” to the manuscript that reflects the limitations of this study and also to add ethical statement of this study at the beginning of the method section.

1- We suggest to include a specific “limitations section” to the manuscript.

Response: At the end of the manuscript we have written the following limitations statement:

“Limitations: The research performed herein is primarily using a live cell platform and a high content microscope within PC3/BSL3 laboratory containment. We recognise that the availability of this infrastructure is limited due to costs and the platform described herein may be restricted. We have demonstrated this system can be modified to be used on a standard plate reader but this still requires appropriate biosafety containment. Furthermore, during COVID-19 surge periods, access to genomic sequencing and also remnant diagnostic material can be delayed for use in research studies. As such, the platforms we describe herein are being embedded in diagnostic laboratories to aid the Covid response.”

2- Also to add ethical statement of this study at the beginning of the method section.

Response: We have moved the following ethics statement to the front of the methods:

“Ethics statement

All human serum samples were obtained with written informed consent from the participants (2020/ETH00964; 2020/ETH02068; 2019/ETH03336; 2021/ETH00180). Primary bronchial epithelial cells (pBEC) were provided by P. A. B. Wark (University of Newcastle), and originally obtained from one healthy non-smoking donor (73-year-old female) during bronchoscopy, with written informed consent. Experiment was conducted with approval from the University of Newcastle Safety Committee (Safety REF# 25/2016 and R5/2017). All subjects underwent fibre-optic bronchoscopy in accordance with standard guideline⁵. All primary isolates used in this study were obtained from de-identified remnant diagnostic swabs that had

completed all diagnostic testing under approval by the New South Wales Chief Health Officer following independent scientific review.”

3- Both HAT24 and HAT10 are HEK293T clones co-expressing ACE-2 and TMPRSS2. What is the reason behind hyper-permissive and sensitive to SARS-CoV-2 infection of HAT24 clone compared to HAT10 clone?

Response: We have looked at early (24 hours) versus latter (72 hour) time points for replication within the HAT10 and HAT24 clones. Early events (entry & fusion) are not significantly different. Where the two clones diverge is at very low levels of virus that can be detected over longer time periods. We have now included this data in the supplementary figure S2 as preliminary observations in understanding the mechanisms of the HAT-24 clone. At present, we are now performing whole genome CRISPR to map what pathways are unique to this cell to give it great sensitivity to SARS-CoV-2 infection.

4- Moreover, the authors mentioned CPE within 8 hours post infection for HAT24 clone in supplementary figure 1. So why has the platform been designed considering only 20 hours of post infection?

Response: Very good question. At the current point in time it has been about work flow within the laboratory and the ability to screen hundreds of serum samples per day. In this setting, the work load is setting up serum dilutions, cells and then plating them over several 384 well plates. The power of the assay after this is one of simplicity. With a set viral dilution (LD50), the virus sustains maximal cytopathic effect in a dose dependent manner after a overnight incubation. The next day, the plate is read on the high content microscope with no further steps.

There are indeed times where reading the sample after 8 hours maybe of priority & the assay can indeed be adapted to a shorter time frame. Furthermore, we are adapting the assay to have a luciferase read-out, where we generate two lines of the HAT24 line with split nanoluciferase. Using this approach, the initial fusion of a small proportion of cells in culture would enable a signal potential well before the 8 hour time point.

6- The authors mentioned that HAT24 clone will be able to isolate SARS-CoV-2 from 80% of PCR positive swabs. What are Ct values of these 20% PCR positive swabs? Is there any relation of virus isolation with high Ct value?

Response: The best example to highlight the answer is the Delta dataset where the swabs are all stored under the same cold chain. In this setting 26% of swabs did not reach titer when the Ct value was between 30 and 38. All Ct values less than 30 were culture positive in this setting (albeit with a continuum of end point titers).

7- In this study, authors reported expansion of B.1.517.1 variant in December 2020 and in July 2021 due to community outbreak in Australia and also calculated Delta to be 9.1 fold more infectious virus than the B.1.517.1. A look at the neutralization antibody response of B.1.517.1 variant among vaccinated and convalescent donors needs to be added? It will be great if this would also include neutralization antibody response of this variant in figure number.

Response: Yes we agree with the reviewer that the addition of this variant to our variant panel would be of interest. During the review of our work we have been asked to include an extensive dataset on Omicron versus the other major VOCs. This data is now presented in figure 7.

Reviewer #2:

Aggarwal and colleagues describe the use of a genetically manipulated cell lines (HAT-24) as a platform for isolating/characterizing SARS-CoV-2 variants. This cell line is highly sensitive to SARS-CoV-2 infection, which allows having readouts from their assays with 24 hours (R-20). Thus, this platform might have a great potential to provide some timely and important results for newly emerge SARS-CoV-2 variants.

This reviewer, however, has some major concerns:

1. The authors actually did not know why HAT-24 is so susceptible to SARS-CoV-2 infection. For this kind of public health studies, it is ok. But the authors should briefly mentioned this in the discussion. In addition, it is not right to say that this cell line “lacked viral restrictions” (Result section). There would be many possible hypotheses to explain this (e.g. increased susceptibility for infection and increase susceptibility to commit cell death).

Response: Yes that is a good point. The priority for our team has been to validate that the cell line can perform in a manner that is sensitive, stringent, reproducible, low in cost, and in high content. This was the priority to ensure we could mobilise this platform in the Covid response. We have performed preliminary assays to determine how the two cells differ and in terms of early cell free entry, we do not observe any differences (See Supplementary Fig. S2). The reviewer is correct in stating we do not know the likely multi-factorial nature of why this clone is more sensitive. We have now changed the results to tone down our interpretation of the HAT-24 line.

“In this setting we introduced a hyper-permissive Hek239T based cell line that co-expresses ACE2 and TMPRSS2 (HAT-24) that was 100 to 1000-fold greater in susceptibility to CPE than the VeroE6 line and approximately ten-fold more than the VeroE6-TMPRSS2 cell line (Fig. S1). Susceptibility was based not only on the expression of physiologically relevant receptors ACE2 and TMPRSS2⁶, but the selection of a clone that was revealed by significant dose dependent cytopathic effect (CPE) accumulating after 8 hours post infection (Movie S1) and one that routinely detected very low levels of virus after 3 days in culture (Fig. S1 and S2). The HEK293T line has limited innate viral immunity, specifically from TAg and adenovirus E1A expression, and this leads to antagonism of antiviral responses by countering interferon (IFN) regulatory factor 3 (IRF3) or IFN-dependent transcription downstream of RNA and DNA sensor activation⁷⁻¹¹. Whilst the latter may contribute to the phenotype, it must be noted it was only one clone that was hyper-permissive to infection (see Fig. S1A-F for a representative clone of 24 (HAT-10 versus HAT-24 in 6 contemporary variants). In this latter setting we recognise the features of this clone may be multifactorial and presently we are using whole genome CRISPR to resolve which pathways are required for this unique cellular phenotype.”

2. *This cell line is unique. As the mechanism of action is unknown, it is impossible for other to generate similar cell lines for the purpose. How can other public health labs can be benefit from this work?*

We are currently depositing the line in global tissue collections and until that is finalised we will readily distribute the cell upon request for all laboratories to benefit. Following the validation herein, this cell line is now used in Australia in diagnostic viral outgrowth assays to clinically inform on the infectious potential of people positive for SARS CoV-2 in various hospital settings.

Yet in the longer term, the mechanistic understanding of why the cell are permissive may shed light in many aspects of virology but also enable *de novo* generation of other cell lines in the future. That said *de novo* genetic engineering, cloning and more importantly validation of each cell/platform takes time and this was the primary aim of our study was to generate and valid platforms to be immediately shared.

3. *The authors use Vero-E6 as a control cell line as a control. This cell line is well known to be not efficient for virus isolation etc. The author actually compared Vero-E6, Vero-E6-TMPRSS2 and HAT-24 side-by-side (Section "Rapid resolution of in vivo....."). It would be more useful if the authors can show the data and show them in the first part of their result section.*

Response: Chronologically we only had access to the Vero-E6-TMPRSS2 line later than the Vero-E6 line. For isolation in pre-Omicron variants we always move from the HAT-24 line to the Vero-E6-TMPRSS2 cell line, as the latter can sustain and produce high viral titers. We have made a note of this here as well as why the utility of the Vero-E6-TMPRSS2 line is important. We have included a representative experiment highlighting the relative CPE per cell line with a variant used throughout this study (B.1.319) in Supp Fig. 1. & have changed the text at the front of the results to highlight this result.

"In this setting we introduced a hyper-permissive Hek239T based cell line that co-expresses ACE2 and TMPRSS2 (HAT-24) that was 100 to 1000-fold greater in susceptibility to CPE than the VeroE6 line and approximately ten-fold more than the VeroE6-TMPRSS2 cell line (Fig. S1)."

4. *Any systematic analysis of the viral replication kinetic in HAT-24 (with high and low MOI)?*

Response: We have included measures of CPE 24hrs versus 72hrs of the HAT-24 versus the representative HAT-10 line in a new supplementary Fig. S2. In addition this is also observed in figure 5 with primary nasopharyngeal swabs, where we have observed infection after 20 versus 96 hours with a significant range of Ct values (ie. Viral loads). The linear regression remains the same for both, but the longer culture period results in greater sensitivity (upward shift of the linear regression) and in Cts greater than 30, it is the longer culture times that enable their detection.

5. Did the authors sequence any virus isolated from this platform? Are these sequences still identical to those from the original clinical samples?

Response: During early viral isolation using this line we have observed no genomic changes (this would be 1 to 4 days in culture post the swab). In our viral stocks do see minor changes (2 to 3 nucleotides) after Vero-e6 or Vero-e6-TMPRSS2 expansions but at present we have not observed these changes in the Spike glycoprotein.

Minor concerns:

1. The Y axis of Fig 2D and 2E should have the range of scale.

Response: Scale bars have now been added.

2. It is bias to cite this preprint (<https://www.medrxiv.org/content/10.1101/2021.09.07.21263229v1.full.pdf>) . There are many studies show that VeroE6-TMPRSS2 can manage to isolate virus from samples with a higher CT value.

Response: we cited this publication as it had cold chain storage of the primary swabs identical to that outlined in this work. Given we have not formally observed this and there will be differences in Ct values across diagnostic networks, we have removed the reference to the VeroE6-TMPRSS2 line here.

3. Figure 6K should be Figure 5K.

Response: This is now corrected

4. Figures 1A, 6A and 6B are not necessary and they can be deleted.

Response: Figure panels 1A, 6A & 6B are now deleted.

5. Figure 6F. What was the time for these cultures, 20 or 96 hrs?

Response: This is now stated as 96 hours in the figure legend.

Reviewer #3 (Remarks to the Author):

Summary: The authors have developed a HEK293T-based cell line for the rapid, efficient rescue of SARS-CoV-2 viral isolates and demonstrated its utility using a large cohort of quarantine surveillance nasopharyngeal swab samples.

This cell line facilitated rapid isolation of SARS-CoV-2 isolates including all relevant variants of concern and interest, from even low titer samples. The authors demonstrated this rapid short term culture platform provided viral end point titer data that correlated well with standard Vero cultures. The platform was used to examine immune evasion of VOCs from a panel convalescent and vaccinee donor sera and further could be used to assess viral infectivity. Using these newly developed approaches, the authors were able to quickly isolate and characterize the Omicron variant as it emerged, demonstrating it to be the most evasive variant to antibody neutralization and confirming Omicron's shift in the mode of fusion/cell entry.

Critique: The study is a very well designed and elegant approach that not only provided a valuable tool for rapid community surveillance, but also was used to generate a very nicely controlled data set for examining viral phenotypes. I have no major critiques, but only minor points for correcting some confusion and perhaps allowing for better communication of certain aspects of the data.

Minor points:

1. Are there other rescue cell lines for SARS-CoV-2 beyond the standard Vero cultures that have been examined/developed? I am not aware of the entirety of this direction of research and it may be worth a couple of sentences in the intro to address this and cite relevant comparators.

Response: Yes that is an excellent point and we have initially added this in the introduction and also clarified this in the results section following the points made by reviewer 2. VeroE6 derived lines are the ones most commonly used and especially those that express TMPRSS2. Whilst our platform enables rapid and unequivocal read-outs over a very short time frame, VeroE6-TMPRSS2 enables the rapid generation of high titer viral stocks for our neutralisation assays. Whilst each platform developed previously has helped in the Covid effort, the platform described herein is unique at many levels. The difference and power of the assay described herein is primarily speed, scale, simplicity and cost. For instance, once the virus is added, the overnight culture self develops in a dose dependent manner & the only thing left to do is read it in a multi-well plate. We have added the following section in the introduction to recognise other powerful platforms. That said, the platform we outline here is unique and used in a manner that is significantly different from the assays we cite here.

“Previous work generating various cell lines and the powerful use of reverse genetic has helped many aspects of the Covid response¹²⁻¹⁴.”

And we have also added the following point Re. VeroE6-TMPRSS2 cells:

“End-point titres using the Spearman-Kärber were calculated at approximately 0.5×10^6 TCID₅₀/ml in VeroE6 and 0.5×10^8 TCID₅₀/ml in the HAT-24 across all 12 variants tested (Fig. 2F and G). Whilst we recognise the VeroE6 line to be relatively insensitive to infection, we further tested the more permissive VeroE6-TMPRSS2 cell line. The relative rank sensitivities of viral titres HAT-24 > VeroE6-TMPRSS2 > VeroE6, with each increase in sensitivity being greater than an order of magnitude (Fig. S1G). Whilst the HAT-24 line can be used in rapid assays, we must note that the cell line VeroE6-TMPRSS2 is required to generate viral stocks of significant titer.”

2. I found the implication that the R-20 platform could accurately assess fitness to be confusing. The authors point out that the rapid rescue with a hyperpermissive cell line has restricted the 'rounds of infection' as compared to the standard Vero cell line. Additionally, as shown with Omicron, the mode of entry can also impact this variable. I may be missing something in how this is communicated, but it seems to me that viral fitness is too multivariable to assess here. The infectivity phenotype data, using PCR Ct values as a readout for comparison is strong, but fitness assessment or the implication that it can be accurately assessed does not seem to be supported or at minimum, communicated well.

Response: we agree that the clarity can be improved in this section. There are two primary issues we see that could provide confusion in how we use the platform to observe viral fitness. The first is that its use changes when detecting viral infectivity as opposed to neutralisation. In this setting the sensitivity can be increased through multiple rounds of viral replication to observe titers of very low-level virus (Cts>30). So it needs to be seen as another powerful use of this platform but in a different context and without a focus on neutralisation (which is better when viral replication is indeed restricted). We have clarified this in the results section by adding the following statement.

“Of note, this use of the platform was primarily for detection of viral titers and importantly at lower viral loads (Ct>30) the longer culture periods and amplification through multiple rounds of infection over several days does diverge from the design of the neutralisation assays described above.”

The second issue is the ability to compare variant fitness. Pre-Omicron, all variants use the same entry pathways and in that setting can be grouped and compared in their ability to utilised that pathway more efficiently (either through ACE2 affinity or better TMPRSS2 use). Omicron is a paradigm shift. Its tropism has changed and so has its receptor use. Only variants that use the same entry pathway can only be fairly compared in this setting (eg. Pre-Omicron Lineages or in contrast lineages of Omicron or variants thereof). The assay did pick up the change in viral entry with a downward shift in the linear regression and we mechanistically resolve this further using with the TMPRSS2 inhibitor Nefamostat. In 2022, it will still be important to rapidly determine the tropism of emerging variants and if they proceed along the same trajectory as Omicron or if they return to the early pool of circulating variants pre-Omicron that efficiently utilised TMPRSS2. The assays herein enable that resolution and can be rapidly updated to express any receptors or cofactors that can track a specific pathway.

We had added the following section to the results to further clarify this point:

“Importantly the R-20 platform revealed its lack of TMPRSS2 use herein and in the future can readily resolve if future SARS-CoV-2 variants either continue along this path in tropism or switch back to TMPRSS2 mediated entry. Importantly at this juncture it must be noted that any measure of viral fitness in any assay will need to consider all pre-Omicron variants as TMPRSS2 dependent and in that setting rank their fitness accordingly. Whilst in Omicron and related lineages that no longer utilise TMPRSS2, the similar grouping based on tropism will need apply prior to any comparisons of relative fitness. This is not unlike other viral families where a switch in tropism results in a paradigm shift on how the virus can or cannot be compared

to other variants. For instance prior to Omicron, ACE2 affinity and increased TMPRSS2 usage defined the increasing fitness of all contemporary SARS-CoV-2 variants in 2021.”

3. Figure 4 labels appear to wrong. Panels G-J state B.1.319 as one of the VOC examined whereas B.1.2 is listed in A-D. These appear inconsistent. Also, in the main text and in the figure legend, it is not immediately clear that the “fold reduction” is comparing. I assume as compared to A2.2, but the text is not clear in this section.

Response: The nomenclature of early variants at times can be confusing and we apologies for the errors here. B.1.319 is the early B clade used in our study and we have ensured this is correct throughout the manuscript. Furthermore, clade A2.2 is the comparator and we had edited the text to clarify this clearly.

4. Main text states that “across all vaccine recipients, breadth was similar across all donors tested”. I am not entirely clear on what this means. I do see some consistencies but the text should be clear about similarities and differences that are being pointed out here.

Response: Yes the precision of this statement could be better. This is now stated as follows:

“Across individual donors, we did observe a range of responses to variants in convalescent versus BNT162b2 vaccine responses. For instance, across all vaccine recipients with titers great than 1/160, 19 had neutralisation breadth across all variants with the exception of Beta (Table S1, S3, S5).”

On behalf on the co-authors of this study, we thank you for the time considering our work

Associate Professor Stuart Turville, Kirby Institute, UNSW Australia

1. Khoury DS, Cromer D, Reynaldi A, et al. Neutralizing antibody levels are highly predictive of immune protection from symptomatic SARS-CoV-2 infection. *Nat Med*. 2021;27(7):1205-1211.
2. Cromer D, Steain M, Reynaldi A, et al. Neutralising antibody titres as predictors of protection against SARS-CoV-2 variants and the impact of boosting: a meta-analysis. *Lancet Microbe*. 2021.
3. Polack FP, Thomas SJ, Kitchin N, et al. Safety and Efficacy of the BNT162b2 mRNA Covid-19 Vaccine. *N Engl J Med*. 2020;383(27):2603-2615.
4. Madhi SA, Ihekweazu C, Rees H, Pollard AJ. Decoupling of omicron variant infections and severe COVID-19. *Lancet*. 2022.
5. Workshop summary and guidelines: investigative use of bronchoscopy, lavage, and bronchial biopsies in asthma and other airway diseases. *J Allergy Clin Immunol*. 1991;88(5):808-814.
6. Hoffmann M, Kleine-Weber H, Schroeder S, et al. SARS-CoV-2 Cell Entry Depends on ACE2 and TMPRSS2 and Is Blocked by a Clinically Proven Protease Inhibitor. *Cell*. 2020;181(2):271-280 e278.
7. Ferreira CB, Sumner RP, Rodriguez-Plata MT, et al. Lentiviral Vector Production Titer Is Not Limited in HEK293T by Induced Intracellular Innate Immunity. *Mol Ther Methods Clin Dev*. 2020;17:209-219.
8. Ronco LV, Karpova AY, Vidal M, Howley PM. Human papillomavirus 16 E6 oncoprotein binds to interferon regulatory factor-3 and inhibits its transcriptional activity. *Genes Dev*. 1998;12(13):2061-2072.
9. Fonseca GJ, Thillainadesan G, Yousef AF, et al. Adenovirus evasion of interferon-mediated innate immunity by direct antagonism of a cellular histone posttranslational modification. *Cell host & microbe*. 2012;11(6):597-606.
10. Bhattacharya S, Eckner R, Grossman S, et al. Cooperation of Stat2 and p300/CBP in signalling induced by interferon-alpha. *Nature*. 1996;383(6598):344-347.
11. Juang YT, Lowther W, Kellum M, et al. Primary activation of interferon A and interferon B gene transcription by interferon regulatory factor 3. *Proceedings of the National Academy of Sciences of the United States of America*. 1998;95(17):9837-9842.
12. Hou YJ, Okuda K, Edwards CE, et al. SARS-CoV-2 Reverse Genetics Reveals a Variable Infection Gradient in the Respiratory Tract. *Cell*. 2020;182(2):429-446 e414.

13. Matsuyama S, Nao N, Shirato K, et al. Enhanced isolation of SARS-CoV-2 by TMPRSS2-expressing cells. *Proceedings of the National Academy of Sciences of the United States of America*. 2020;117(13):7001-7003.
14. Planas D, Bruel T, Grzelak L, et al. Sensitivity of infectious SARS-CoV-2 B.1.1.7 and B.1.351 variants to neutralizing antibodies. *Nat Med*. 2021;27(5):917-924.

Decision Letter, first revision:

Our ref: NMICROBIOL-21123138A

14th March 2022

Dear Stuart,

Thank you for submitting your revised manuscript "Rapid isolation and resolution immune evasion and viral fitness across contemporary SARS-CoV-2 variants" (NMICROBIOL-21123138A). It has now been seen by the original referees and their comments are below. The reviewers find that the paper has improved in revision, and therefore we'll be happy in principle to publish it in Nature Microbiology, pending minor revisions to comply with our editorial and formatting guidelines.

Thank you again for your interest in Nature Microbiology. Please do not hesitate to contact me if you have any questions.

Sincerely,

[Redacted]

Reviewer #1 (Remarks to the Author):

In my opinion the acceptance of this manuscript for publication

Reviewer #2 (Remarks to the Author):

Thanks for addressing the comments.

Reviewer #4 (Remarks to the Author):

I greatly appreciate the authors providing more thorough details on their VE estimates as well as highlighting the limitations of these estimates. I also appreciate that the authors provided more discussion on the lack of neutralizing antibodies does not necessarily mean lack of protection. The authors have adequately addressed my concerns.

Decision Letter, Final Checks:

Our ref: NMICROBIOL-21123138A

22nd March 2022

Dear Stuart,

Thank you for your patience as we've prepared the guidelines for final submission of your Nature Microbiology manuscript, "Rapid isolation and resolution immune evasion and viral fitness across contemporary SARS-CoV-2 variants" (NMICROBIOL-21123138A). Please carefully follow the step-by-step instructions provided in the attached file, and add a response in each row of the table to indicate the changes that you have made. Please also check and comment on any additional marked-up edits we have proposed within the text. Ensuring that each point is addressed will help to ensure that your revised manuscript can be swiftly handed over to our production team.

We would like to start working on your revised paper, with all of the requested files and forms, as soon as possible (preferably within two weeks or ideally faster given the topic). Please get in contact with us if you anticipate delays.

In recognition of the time and expertise our reviewers provide to Nature Microbiology's editorial process, we would like to formally acknowledge their contribution to the external peer review of your manuscript entitled "Rapid isolation and resolution immune evasion and viral fitness across contemporary SARS-CoV-2 variants". For those reviewers who give their assent, we will be publishing their names alongside the published article.

Nature Microbiology offers a Transparent Peer Review option for new original research manuscripts

submitted after December 1st, 2019. As part of this initiative, we encourage our authors to support increased transparency into the peer review process by agreeing to have the reviewer comments, author rebuttal letters, and editorial decision letters published as a Supplementary item. When you submit your final files please clearly state in your cover letter whether or not you would like to participate in this initiative. Please note that failure to state your preference will result in delays in accepting your manuscript for publication.

Cover suggestions

As you prepare your final files we encourage you to consider whether you have any images or illustrations that may be appropriate for use on the cover of Nature Microbiology.

Nature Microbiology has now transitioned to a unified Rights Collection system which will allow our Author Services team to quickly and easily collect the rights and permissions required to publish your work. Approximately 10 days after your paper is formally accepted, you will receive an email in providing you with a link to complete the grant of rights. If your paper is eligible for Open Access, our Author Services team will also be in touch regarding any additional information that may be required to arrange payment for your article.

Please note that *Nature Microbiology* is a Transformative Journal (TJ). Authors may publish their research with us through the traditional subscription access route or make their paper immediately open access through payment of an article-processing charge (APC). Authors will not be required to make a final decision about access to their article until it has been accepted. [Find out more about Transformative Journals](https://www.springernature.com/gp/open-research/transformative-journals)

Authors may need to take specific actions to achieve [compliance](https://www.springernature.com/gp/open-research/funding/policy-compliance-faqs) with funder and institutional open access mandates. If your research is supported by a funder that requires immediate open access (e.g. according to [Plan S principles](https://www.springernature.com/gp/open-research/plan-s-compliance))

then you should select the gold OA route, and we will direct you to the compliant route where possible. For authors selecting the subscription publication route, the journal's standard licensing terms will need to be accepted, including [self-archiving policies](https://www.springernature.com/gp/open-research/policies/journal-policies). Those licensing terms will supersede any other terms that the author or any third party may assert apply to any version of the manuscript.

Please use the following link for uploading these materials:
[Redacted]

Best regards,

[Redacted]

Reviewer #1:
Remarks to the Author:
In my opinion the acceptance of this manuscript for publication

Reviewer #2:
Remarks to the Author:
Thanks for addressing the comments.

Reviewer #4:
Remarks to the Author:
I greatly appreciate the authors providing more thorough details on their VE estimates as well as highlighting the limitations of these estimates. I also appreciate that the authors provided more discussion on the lack of neutralizing antibodies does not necessarily mean lack of protection. The authors have adequately addressed my concerns.

Final Decision Letter:

Dear Stuart,

I am pleased to accept your Article "Platform for isolation and characterization of SARS-CoV-2 variants enables rapid characterisation of Omicron in Australia" for publication in Nature Microbiology and thank

you for sending back your revisions so promptly. Thank you for having chosen to submit your work to us and many congratulations.

Due to the importance of these deadlines, we ask you to please let us know now whether you will be difficult to contact over the next month. If this is the case, we ask you to provide us with the contact information (email, phone and fax) of someone who will be able to check the proofs on your behalf, and who will be available to address any last-minute problems.

Acceptance of your manuscript is conditional on all authors' agreement with our publication policies (see <https://www.nature.com/nmicrobiol/editorial-policies>). In particular your manuscript must not be published elsewhere and there must be no announcement of the work to any media outlet until the publication date (the day on which it is uploaded onto our website).

Please note that *Nature Microbiology* is a Transformative Journal (TJ). Authors may publish their research with us through the traditional subscription access route or make their paper immediately open access through payment of an article-processing charge (APC). Authors will not be required to make a final decision about access to their article until it has been accepted. [Find out more about Transformative Journals](https://www.springernature.com/gp/open-research/transformative-journals)

Authors may need to take specific actions to achieve [compliance with funder and institutional open access mandates](https://www.springernature.com/gp/open-research/funding/policy-compliance-faqs). If your research is supported by a funder that requires immediate open access (e.g. according to [Plan S principles](https://www.springernature.com/gp/open-research/plan-s-compliance)) then you should select the gold OA route, and we will direct you to the compliant route where possible. For authors selecting the subscription publication route, the journal's standard licensing terms will need to be accepted, including [self-archiving policies](https://www.nature.com/nature-portfolio/editorial-policies/self-archiving-and-license-to-publish). Those licensing terms will supersede any other terms that the author or any third party may assert apply to any version of the manuscript.

We welcome the submission of potential cover material (including a short caption of around 40 words) related to your manuscript; suggestions should be sent to Nature Microbiology as electronic files (the

image should be 300 dpi at 210 x 297 mm in either TIFF or JPEG format). Please note that such pictures should be selected more for their aesthetic appeal than for their scientific content, and that colour images work better than black and white or grayscale images. Please do not try to design a cover with the Nature Microbiology logo etc., and please do not submit composites of images related to your work. I am sure you will understand that we cannot make any promise as to whether any of your suggestions might be selected for the cover of the journal.
